# Texture or Semantics? Vision-Language Models Get Lost in Font Recognition

**Zhecheng Li**[†]   **Guoxian Song**[∥]   **Yujun Cai**[§]   **Zhen Xiong**[δ]
**Junsong Yuan**[⋆]   **Yiwei Wang**[‡]
[†] University of California, San Diego   [∥] ByteDance
[§] The University of Queensland   [δ] University of Southern California
[⋆] University at Buffalo   [‡] University of California, Merced
zhl186@ucsd.edu
https://github.com/Lizhecheng02/VLM4Font

## Abstract

Modern Vision-Language Models (VLMs) exhibit remarkable visual and linguistic capabilities, achieving impressive performance in various tasks such as image recognition and object localization. However, their effectiveness in fine-grained tasks remains an open question. In everyday scenarios, individuals encountering design materials, such as magazines, typography tutorials, research papers, or branding content, may wish to identify aesthetically pleasing fonts used in the text. Given their multimodal capabilities and free accessibility, many VLMs are often considered potential tools for font recognition. This raises a fundamental question: ***Do VLMs truly possess the capability to recognize fonts?*** To investigate this, we introduce the *Font Recognition Benchmark (FRB)*, a compact and well-structured dataset comprising 15 commonly used fonts. FRB includes two versions: (i) *an easy version*, where 10 sentences are rendered in different fonts, and (ii) *a hard version*, where each text sample consists of the names of the 15 fonts themselves, introducing a stroop effect that challenges model perception. Through extensive evaluation of various VLMs on font recognition tasks, we arrive at the following key findings: (i) Current VLMs exhibit limited font recognition capabilities, with many state-of-the-art models failing to achieve satisfactory performance and being easily affected by the stroop effect introduced by textual information. (ii) Few-shot learning and Chain-of-Thought (CoT) prompting provide minimal benefits in improving font recognition accuracy across different VLMs. (iii) Attention analysis sheds light on the inherent limitations of VLMs in capturing semantic features.

## 1 Introduction

Font recognition is a fundamental visual task with applications across various domains. It is particularly relevant to graphic design, publishing, advertising, and digital content creation, where typography plays a crucial role in visual communication (Zramdini & Ingold, 1998; Chen et al., 2014b; Zhu et al., 2001). For example, researchers may seek to identify aesthetically appealing fonts in academic papers or images, while designers sometimes encounter unique typefaces they wish to integrate into their projects. Statistics indicate that over 60% of websites utilize custom or non-system fonts, highlighting the significance of font identification in web design. Additionally, Google Fonts are used on more than 50 million websites, demonstrating the extensive adoption of diverse typefaces (Solomons, 2023). This widespread demand underscores the need for efficient and accurate font recognition tools.

Traditionally, Convolutional Neural Networks (CNNs) have been the predominant architecture for font recognition (Chen et al., 2021; Li et al., 2022b). While CNNs achieve high accuracy, they rely on large and well-annotated datasets for training (Wang & Zong, 2023; Zhang, 2023; Tonmoy et al., 2024). The process of data collection, annotation, and

curation is time-consuming and resource-intensive, limiting the scalability and adaptability of CNN-based systems in real-world applications. In contrast, VLMs, pre-trained on large-scale multimodal datasets, have demonstrated remarkable capabilities in processing both visual and textual information (Zhang et al., 2024c; Bordes et al., 2024; OpenAI et al., 2024b; OpenAI, 2024; Team et al., 2024; Google, 2025). Their ability to generalize across diverse visual tasks without extensive task-specific training makes them highly versatile. Additionally, many VLMs are open-weight and freely accessible, positioning them as practical alternatives for everyday font recognition.

Although VLMs have the potential to aid in font recognition, there has been no comprehensive analysis or systematic evaluation of their capabilities in this domain. In this paper, we introduce a font recognition benchmark comprising 15 commonly used fonts, with images containing textual content rendered in these typefaces. To ensure a rigorous evaluation, we construct two dataset

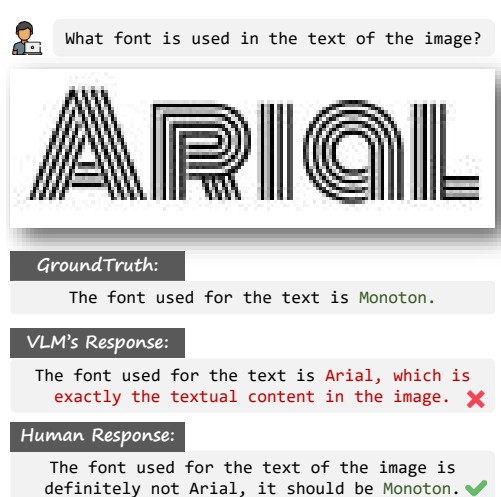

Figure 1: An illustrative example where the image displays the word 'Arial' rendered in the 'Monoton' font. The vision-language model incorrectly predicts the font as 'Arial', indicating an over-reliance on textual content and limited sensitivity to font semantics.

versions: an easy version, where fonts are recognized from images containing sentences, and a hard version, which introduces the stroop effect (Besner et al., 1997; MacLeod, 1991; Verhaeghen & De Meersman, 1998) by displaying font names in mismatched typefaces, adding an additional layer of difficulty. This benchmark provides a systematic assessment of state-of-the-art VLMs' font recognition abilities.

We conduct extensive evaluations on 13 open-weight and closed-source VLMs. All tested models are struggle to differentiate between font types and exhibit high sensitivity to the textual content within images, as illustrated in Figure 1. The best-performing model achieves only around 30% accuracy on the easy version of the benchmark, while performance on the hard version is even more challenging, with accuracy dropping to approximately 15%. Additionally, chain-of-thought (CoT) prompting (Wei et al., 2023; Kojima et al., 2023) provides limited performance gains in this task. Furthermore, we incorporate single-letter images as demonstrations in few-shot learning scenarios to help VLMs identify font similarities. Interestingly, despite the widespread familiarity of these fonts and the relatively low difficulty of the task for humans, where few-shot examples significantly aid in font recognition, current VLMs continue to exhibit poor performance.

In summary, our contributions are threefold:

(i) We propose a font recognition benchmark with two versions to systematically evaluate the capabilities of VLMs and conduct a comprehensive assessment of 13 different models.

(ii) Our results reveal that current VLMs perform poorly in font recognition, with neither Chain-of-Thought prompting nor few-shot learning providing significant improvements.

(iii) We analyze attention matrices across image patches to gain deeper insights into their failure modes and susceptibility to texture-based perturbations.

## 2 Related Works

### 2.1 Vision-Language Models

With the foundation laid by Transformer and CLIP (Vaswani et al., 2023; Radford et al., 2021), Vision-Language Models (VLMs) have emerged as powerful multimodal systems that

seamlessly integrate both visual and linguistic modalities. These models bridges the gap between image and text understanding, enabling them to process and interpret complex multimodal data efficiently (OpenAI et al., 2024a; Bai et al., 2025; Team et al., 2024; Li et al., 2024; Liu et al., 2023; Caffagni et al., 2024; Zhang et al., 2024a; Yin et al., 2024; Team et al., 2025; Chen et al., 2024).

Recent advancements in large-scale image-text datasets and contrastive learning have significantly improved VLM performance. These models now exhibit strong general representation capabilities, effectively aligning visual and textual information and achieving state-of-the-art results across multiple benchmarks, establishing themselves as key technologies in multimodal AI research (Zhang et al., 2024c; Bordes et al., 2024; Schuhmann et al., 2022; Li et al., 2023; Chen et al., 2025; 2020).

The strong multimodal capabilities of VLMs make them valuable tools in real-world applications. Models such as BLIP (Li et al., 2022a), Flamingo (Alayrac et al., 2022) and later GPT-4O (OpenAI et al., 2024b) exhibit impressive performance in visual question answering, image captioning, and multimodal dialogue, enabling applications in interactive AI agents and assistive technologies for visually impaired individuals (Agrawal et al., 2016; Zhang et al., 2024b; Guo et al., 2024; Gao et al., 2024; Zhai et al., 2024). These advancements underscore the versatility of VLMs in handling complex multimodal tasks.

## 2.2 Font Recognition

Font recognition, a critical task for people such as researchers and designers, inherently combines visual and textual components. However, it has received limited attention within the Vision-Language Models (VLMs) research community. Traditionally, it has been approached as a text-image classification problem, predominantly relying on CNN-based methods (Wang et al., 2015b; Tensmeyer et al., 2017; Qi et al., 2025; Wang & Zong, 2023; Mohammadian et al., 2022; Cui & Inoue, 2021). While these approaches achieve relatively high accuracy with their proposed CNN-based architectures on specific datasets, they typically require extensive fine-tuning on large-scale datasets. This requirement makes them time-consuming and also limits their generalization capabilities.

VLMs have recently emerged as versatile tools in visual tasks, offering strong adaptability through large-scale pretraining. They present a promising, convenient solution for font recognition. However, their capabilities in this task remain underexplored, motivating our study to provide a systematic evaluation and identify potential limitations.

## 3 Font Recognition Benchmark

We propose the Font Recognition Benchmark (FRB) to assess VLMs' ability to recognize typographic styles from images. Unlike different versions or subsets of the VFR dataset, which contains more than 2,400 font classes and further categorizes the same font style into regular, bold, and other sub-types, the VFR dataset is more suitable for CNN-based model training rather than for basic capability exploration of VLMs in line with our objectives (Wang et al., 2015a; Chen et al., 2014a). In contrast, FRB follows three principles: practicality, using commonly encountered fonts; diversity, covering multiple font categories; and graduated challenge, testing different recognition scenarios. It includes two versions: an easy version, where sentences appear in various fonts, and a hard version, where font names are displayed in mismatched typefaces, creating a stroop effect.

### 3.1 Font Selection

Fonts are generally categorized into three major types: Serif Fonts, Sans-Serif Fonts, and Script & Decorative Fonts. To ensure stylistic diversity in our experiments, we select 15 widely used fonts from these three categories, We choose these fonts based on responses from GPT-4O, CLAUDE-3.5-SONNET, and statistical data (as shown in Appendix A). The classification and corresponding font names are as follows:

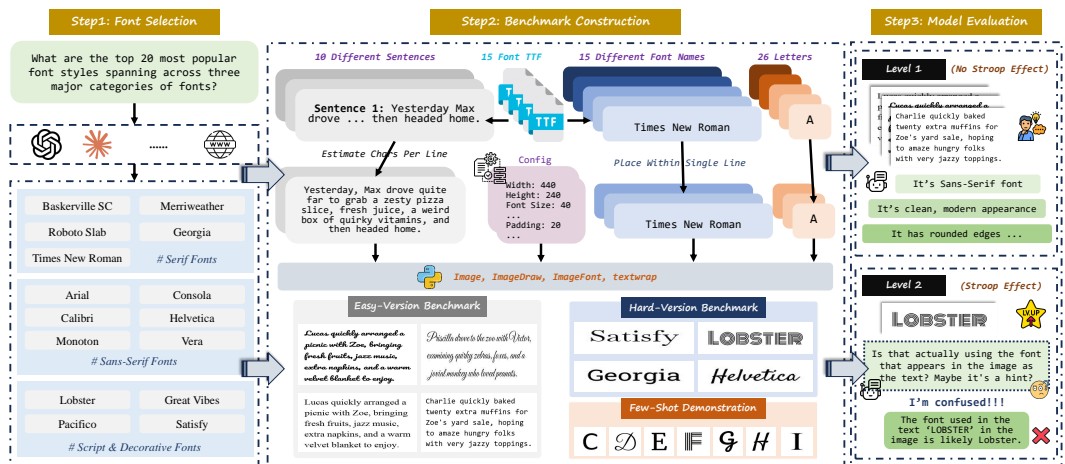

Figure 2: An overview of the complete pipeline for constructing the font recognition benchmark, including the two-tier evaluation design. TTF files and corresponding codes ensure visual consistency across images. The easy task evaluates the basic font recognition ability of VLMs, while the hard task introduces stroop effect to assess robustness under semantic-visual conflict.

**Serif Fonts:** Baskerville SC, Georgia, Merriweather, Times New Roman, Roboto Slab.
**Sans-Serif Fonts:** Arial, Calibri, Consola, Helvetica, Monoton, Vera.
**Script & Decorative Fonts:** Great Vibes, Lobster, Pacifico, Satisfy.

## 3.2 Benchmark Image Construction

We generate 10 images per font, each containing a distinct sentence designed to incorporate as many of the 26 English letters as possible. This results in a total of 15 × 10 = 150 images, forming the easy version of the benchmark. These images are created using a consistent Python script that ensures uniform line breaks, font size, and background, maintaining a balanced aspect ratio aligned with the model's input requirements. This easy version allows VLMs to focus exclusively on font recognition without interference from textual content.

For the hard version of the benchmark, we generate 15 additional images per font, each displaying its respective name. The image generation process employs a unified Python script utilizing the *ImageDraw* and *ImageFont* packages, ensuring consistency in font size and background dimensions across all images. This approach eliminates discrepancies associated with alternative methods such as screenshots. The inclusion of font names introduces the stroop effect, increasing the benchmark's difficulty. In total, this results in 15 × 15 = 225 images, constituting the complete hard version of the benchmark.

Finally, to support few-shot learning experiments, we generate 52 additional images, each containing one uppercase or lowercase letter from the 26 English alphabets. These images serve as reference samples, allowing VLMs to leverage few-shot learning for feature comparison and adaptation in subsequent experiments.

## 3.3 Challenges

### 3.3.1 Font Recognition

When the text content in an image consists of sentences, this task simulates real-world scenarios where individuals capture screenshots of textual content to identify the font used. The primary objective is to assess the VLMs' ability to recognize fonts. The presence of a greater number of letters enhances the models' capacity to capture distinctive font characteristics, while the relatively small text size further challenges its ability to perform fine-grained recognition.

| Base Model | EASY-VERSION | | | | HARD-VERSION | | | |
|---|---|---|---|---|---|---|---|---|
| | Zero-Shot | Zero-Shot CoT | Zero-Shot$^M$ | Zero-Shot CoT$^M$ | Zero-Shot | Zero-Shot CoT | Zero-Shot$^M$ | Zero-Shot CoT$^M$ |
| **Closed-Source Models** | | | | | | | | |
| GPT-4O | 24.67 | 30.67 | **66.67** | **46.67** | 12.44 | **15.11** | **27.11** | **25.78** |
| CLAUDE-3.5-SONNET | **31.33** | 29.33 | 28.67 | 29.33 | 8.44 | 7.11 | 7.11 | 6.67 |
| GEMINI-2.0-FLASH-001 | 18.67 | 18.00 | 18.67 | 18.00 | 9.78 | 9.78 | 9.78 | 9.78 |
| GPT-4O-MINI | 0.00 | 2.67 | 33.33 | 28.00 | 4.44 | 4.89 | 8.44 | 8.89 |
| **Open-Weight Models** | | | | | | | | |
| LLAMA-3.2-90B-VISION-INSTRUCT | 9.33 | 12.67 | 11.33 | 11.33 | 8.44 | 9.78 | 9.33 | 7.11 |
| LLAMA-3.2-11B-VISION-INSTRUCT | 18.67 | 9.33 | 16.67 | 14.00 | **13.33** | 7.56 | 6.67 | 5.33 |
| PHI-3.5-VISION-INSTRUCT | 4.67 | 0.00 | 6.67 | 3.33 | 2.67 | 2.22 | 6.67 | 6.22 |
| PHI-3-VISION-128K-INSTRUCT | 0.67 | 0.00 | 10.00 | 4.67 | 0.00 | 0.89 | 6.67 | 5.33 |
| QWEN2-VL-7B-INSTRUCT | 0.00 | 2.00 | 7.33 | 7.33 | 4.44 | 4.44 | 6.67 | 7.56 |
| QWEN2.5-VL-7B-INSTRUCT | 12.00 | 5.33 | 17.33 | 12.00 | 5.33 | 5.33 | 6.67 | 8.00 |
| QWEN2-VL-72B-INSTRUCT | 8.00 | 7.33 | 6.67 | 7.33 | 6.22 | 6.22 | 6.67 | 5.78 |
| IDEFICS3-8B-LLAMA3 | 1.33 | 1.33 | 7.33 | 7.33 | 4.89 | 0.89 | 6.67 | 6.22 |
| IDEFICS2-8B | 8.00 | 11.33 | 9.33 | 8.00 | 8.44 | 7.11 | 6.22 | 6.22 |

Table 1: The accuracy of 13 open-weight and closed-source vision-language models on both the easy and hard versions of the benchmark under different settings. $^M$ indicates the use of a MCQ setting for inference. The highest accuracy in each setting is highlighted in **bold**, while the second-best results are underlined.

### 3.3.2  *Stroop Effect*

When image text contains font names, a conflict between the textual meaning and the visual font triggers the stroop effect (MacLeod, 1991; Verhaeghen & De Meersman, 1998; Besner et al., 1997). For instance, the word "Times New Roman" rendered in **Lobster** font may mislead a model to predict the text's meaning rather than the actual font. Even with clear visual differences, models often misclassify due to cognitive interference. This setup challenges VLMs to disentangle textual semantics from typographic appearance.

## 4  Experimental Setup

### 4.1  Vision Language Models

To effectively assess the capabilities of the latest VLMs, we conduct experiments on 13 widely recognized models. Detailed information about these models is provided in Appendix B.

### 4.2  Few-Shot Selection

Given the large number of tokens required for image input and the common real-world practice of identifying fonts by analyzing individual letter styles, we construct a dataset comprising 52 images, each representing an uppercase or lowercase letter from the 26 English alphabets. To ensure consistency and fairness in selecting few-shot samples, we employ the CLIP model for image similarity retrieval. In this setup, the 52 single-letter images serve as the retrieval database, while the experimental images containing font names and sentences function as target samples. We design six experimental scenarios, ranging from 1-shot to 6-shot settings.

## 5  Experimental Results

We evaluate multiple VLMs using our proposed font recognition benchmark. The inference methods include the standard zero-shot approach, as well as Chain-of-Thought (CoT) (Wei et al., 2023; Kojima et al., 2023) and few-shot methods, providing a comprehensive assess-

ment of VLMs' font recognition capabilities. All experimental prompts are provided in Appendix F.

## 5.1 Vision-Language Models Fail on Font Recognition Task

Table 1 demonstrates that contemporary VLMs exhibit suboptimal performance on font recognition tasks, despite the selected 15 fonts being among the more commonly used ones across three major font categories. Under the easy version of the benchmark in a zero-shot setting, the CLAUDE-3.5-SONNET achieves the highest accuracy among the evaluated models but only reaches approximately 31%, while powerful GPT-4O-MINI achieves even 0% accuracy. Among open-weight models, performance remains notably poor, with only LLAMA-3.2-11B-VISION-INSTRUCT surpassing 10%, achieving an accuracy of nearly 19%.

Performance further deteriorates under the hard version of the benchmark, which incorporates the stroop effect. In this setting, almost all models fail to exceed 15% accuracy in a zero-shot scenario. The sole exception is GPT-4O, which attains around 15% accuracy while using zero-shot CoT. Even other powerful closed-source models fail to surpass 10% accuracy, regardless of whether CoT is applied. These findings underscore the fundamental limitations of existing VLMs in font recognition tasks.

## 5.2 Vision-Language Models Can Rely on Spurious Visual Cues

To further investigate whether contemporary VLMs possess font recognition capabilities, we conduct an experiment using multiple-choice questions (MCQ) under zero-shot setting, where models are explicitly instructed to select the correct font category from the 15 given options. The results of this experiment are also presented in Table 1. We now compare these experimental results under the setting that does not use CoT prompting:

For the easy version of the benchmark, we observe that among the four powerful closed-source models, MCQ does not lead to a significant improvement for CLAUDE-3.5-SONNET and GEMINI-2.0-FLASH-001. However, it significantly improves the accuracy of the two GPT-series models. Among all open-weight models, MCQ improves accuracy by 6–10 percentage points for

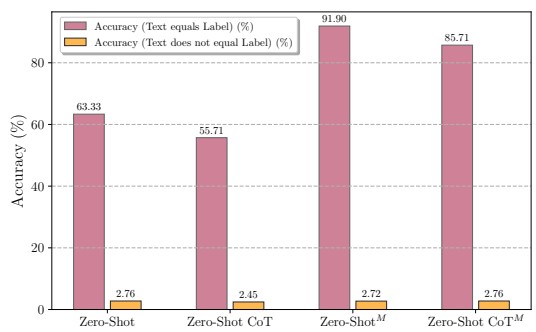

Figure 3: Accuracy comparison on the hard version of the benchmark under two labeling conditions: whether the rendered content in the image conflicts with the actual font label. M denotes inference under a multiple-choice setting.

IDEFICS3-8B-LLAMA3, QWEN2-VL-7B-INSTRUCT, and PHI-3-VISION-128K-INSTRUCT. However, this improvement remains marginal, as their original accuracy is close to 0%, indicating that even with a constrained choice set, these models still perform poorly. For other models, the accuracy gains under the MCQ setting are smaller, remaining below 4 percentage points.

For the hard version of the benchmark, among closed-source models, only GPT-4O exhibits a significant improvement in accuracy under the MCQ setting, while other models show little to no change. Among open-weight models, a notable finding is that after applying MCQ, all models achieve an accuracy close to 6.67% — precisely the expected accuracy if a model correctly identifies the font only when the font style matches the font name in the image. Figure 3 demonstrates that applying MCQ to the hard version of the benchmark only further amplifies the influence of the stroop effect. However, it has minimal impact when the text in the image does not match its font type. This strongly suggests that these models lack the capability to discern the correct font under the stroop effect; instead, they are merely influenced by the textual content in the image. This result further underscores the susceptibility of VLMs to the stroop effect in this task. Given their inherent limitations in font

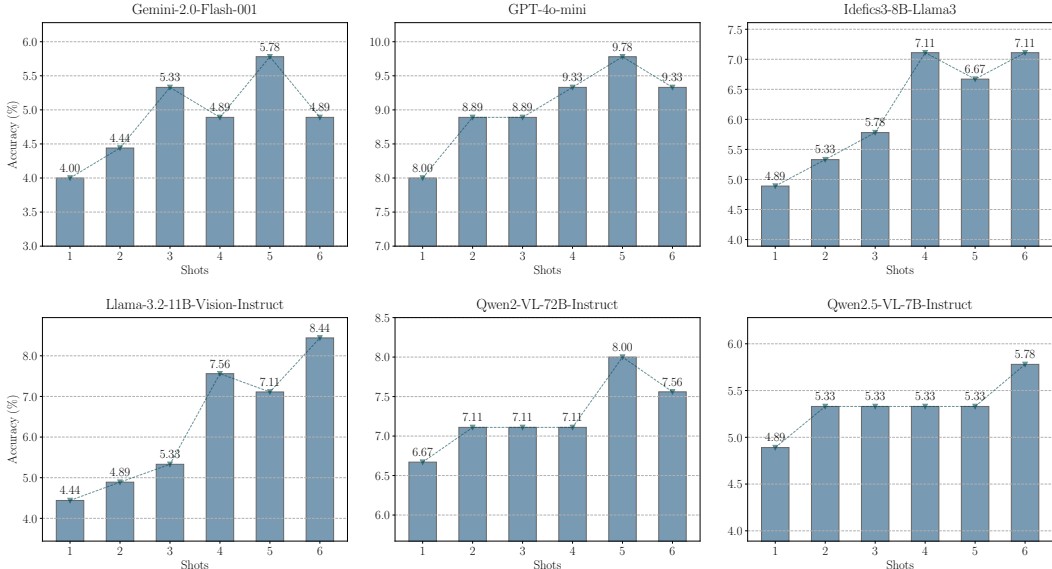

Figure 4: Few-shot accuracy (1-shot to 6-shot) on the hard benchmark setting, evaluated across six vision-language models from distinct model families.

recognition, even when provided with explicit answer choices, these models fail to leverage the constraints meaningfully. Instead, they appear to be even more influenced by the textual content, as evidenced by the accuracy drop observed in LLAMA-3.2-11B-VISION-INSTRUCT on the hard version.

### 5.3   Few-shot Learning Provides Limited Improvements

In real-world scenarios, when given reference images of individual letters written in a specific font style, humans can carefully compare these samples with a test image to determine the font type. This raises the question of whether few-shot learning can enhance VLMs' font recognition capabilities — specifically, whether VLMs can focus on the features provided in the demonstrations and leverage comparison-based reasoning to identify the correct font in the final test image.

To investigate this, we conduct few-shot experiments on six open-weight and closed-source VLMs under the setting described in Section 4.2. As shown in Figure 4, increasing the number of shots from 1 to 6 results in only a marginal improvement in model accuracy, with gains of less than 4 percentage points across all models. Given the dataset size for the hard version of the benchmark evaluation, this level of improvement is not ideal. Furthermore, we observe that, with the exception of the QWEN2-VL-72B-INSTRUCT, whose accuracy either increases or remains stable as the number of shots increases, other models exhibit fluctuations in accuracy despite receiving more examples, which is counterintuitive. This instability further underscores the limited capabilities of VLMs in font recognition under few-shot settings. Notably, this contrasts sharply with human performance in fine-grained recognition tasks, where individuals effectively leverage reference samples to improve accuracy.

## 6   Categories of Errors

To investigate the reasons behind the suboptimal performance of VLMs in font recognition, we conduct a detailed analysis of model outputs to identify specific error patterns. Based on our observations, we classify the errors into four distinct categories and provide representative examples of each in Table 3.

| Error Type | Easy Version (%) | Hard Version (%) |
|---|---|---|
| Content-Biased Misattribution | 0.00 | 70.11 |
| Explicit Inability Acknowledgment | 10.88 | 2.13 |
| Indecisive Classification | 64.87 | 14.46 |
| Confidently Incorrect Prediction | 24.25 | 13.30 |

Table 2: Error ratios for four distinct error categories, aggregated over all incorrect predictions by 13 vision-language models on both benchmark tiers (easy and hard).

**Content-Biased Misattribution:** Error occurs primarily in the hard version of the benchmark, where the model mistakenly assumes that the font name appearing in the image corresponds to the actual font used for the text.

**Explicit Inability Acknowledgment:** The model explicitly admits it cannot determine the font, often stating *"more information is needed"* or *"I can't determine the font"*, reflecting awareness of its limitations.

**Indecisive Classification:** The model analyzes font characteristics and distinguishes broad categories like Serif vs. Sans-Serif but fails to name a specific font, instead listing multiple candidates.

**Confidently Incorrect Prediction:** The model gives a definitive but incorrect answer, sometimes with font analysis, displaying unjustified confidence and leading to misleading results.

Table 2 presents the distribution of error types across two benchmark levels. Notably, in the easy version, where the image content consists of complete sentences, content-biased misattribution does not occur. Although the textual content in this setting does not pose a stroop effect challenge, a considerable portion of the model responses indicate either an explicit acknowledgment of their inability to perform the font recognition task or the provision of only a set of possible candidates or a broader font category, without a definitive prediction. This suggests that current VLMs lack the robustness required for accurate font recognition and often exhibit uncertainty when faced with this task.

In contrast, in the hard version, the presence of font names within the images introduces significant confusion. Even when utilizing Chain-of-Thought prompting, most models struggle to consistently analyze font features and instead rely heavily on the textual content. Only a few sufficiently strong models demonstrate the ability to reason in the correct direction. These findings indicate that current VLMs lack robustness and are not yet capable of reliably handling font recognition tasks.

## 7 Do VLMs Truly Focus on Crucial Edge Regions?

Different VLMs utilize distinct mechanisms for processing image inputs. Considering both their popularity and open-weight availability, we analyze the attention matrices from two models: LLAMA-3.2-11B-VISION-INSTRUCT and QWEN2-VL-7B-INSTRUCT.

Figure 5 presents the attention matrix of the final cross-attention layer for the LLAMA-3.2-11B-VISION-INSTRUCT model on incorrect predictions from the hard version of the benchmark. All images in this figure share a common characteristic: the displayed font name does not match the actual font used. If VLMs truly possess the ability to identify fonts, their predictions should not be heavily influenced by this discrepancy. For instance, a human observer would easily recognize that the text displaying 'Baskervville SC' is not written in the Baskervville SC font, as the characteristics of the letters suggest a script-style font. However, as observed in the attention matrix, LLAMA-3.2-11B-VISION-INSTRUCT fails to focus on the regions that capture distinctive font features, such as the edges of individual characters. Instead, across all six sub-images, the model exhibits misplaced attention, often focusing on irrelevant background regions rather than the critical font-defining elements. This misalignment in attention leads to the failure of the recognition task, highlighting the model's limitations in accurately distinguishing font styles.

| Error Category | Label | Incorrect Model Prediction |
|---|---|---|
| Content-Biased Misattribution | Lobster | The text in the image is 'Georgia'. This is a font name, which is used to describe the style and appearance of the text. |
| | Consola | The image contains the text 'calibri'. The font used in the text is Calibri. |
| Explicit Inability Acknowledgment | Calibri | The information provided is not sufficient for me. Therefore, it is not possible to determine the font used in the text of the image. |
| | Arial | I'm not able to identify the font used in the image. If you have access to font-identification tools, they might provide a more accurate result. |
| Indecisive Classification | Arial | The font is likely a standard system font, such as Calibri or Helvetica, which are widely used in various applications and documents. |
| | Great Vibes | Based on the characteristics, it seems to resemble a font like Dancing Script or Pacifico, but it's hard to pinpoint the exact name. |
| Confidently Incorrect Prediction | Arial | Based on these observations, the font resembles Georgia. Georgia is known for its readability on screens and has a classic serif style with noticeable stroke contrast. |
| | Helvetica | Based on these observations, the font resembles Times New Roman. This font is widely used for its readability and classic style. |

Table 3: Examples of four different error types in both the easy and hard versions of the benchmark, displaying the incorrect model predictions alongside the corresponding ground-truth labels.

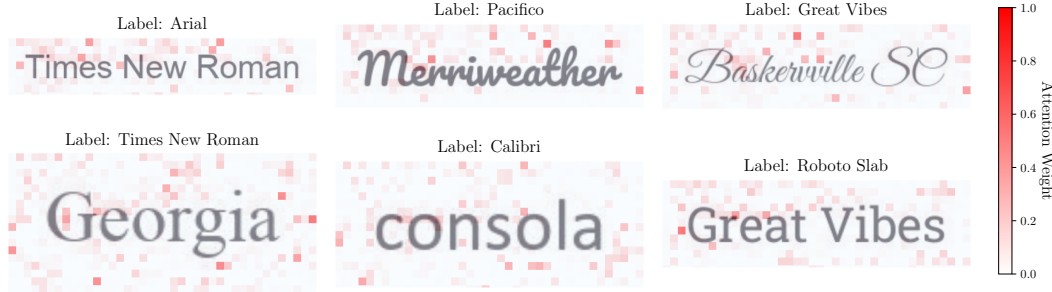

Figure 5: Attention heatmaps from the final cross-attention layer of LLAMA-3.2-11B-VISION-INSTRUCT, illustrating how text tokens attend to image patches for six input images from the hard version of the benchmark, all of which the model fails to classify correctly. Attention weights are averaged across all heads.

Figure 6 presents the attention matrix for QWEN2-VL-7B-INSTRUCT on the easy version of the benchmark. Compared to LLAMA-3.2-11B-VISION-INSTRUCT, QWEN2-VL-7B-INSTRUCT employs a larger merged patch size. However, regardless of patch size, a similar issue persists: the model's attention does not sufficiently focus on the edges of the letters in the image. Instead, only a few letters receive more attention than other parts, and in many cases, the model assigns higher attention to the white background rather than the textual content. However, when comparing the attention matrices in Figure 6 with those in Figure 5, we observe that in the easy version of the benchmark, QWEN2-VL-7B-INSTRUCT demonstrates a stronger tendency to focus on the textual regions. The red-highlighted areas in the attention matrix cover most of the text, even though the highest attention weights are not fully concentrated on the letters. Nevertheless, this represents a notable improvement over LLAMA-3.2-11B-VISION-INSTRUCT 's performance on the hard version of the benchmark, where attention is more poorly distributed.

Based on our analysis of the above two VLMs, we infer that current VLMs remain inadequate for font recognition tasks, even for widely used font styles. This limitation likely arises from their inability to focus on the critical regions of the textual input that are essential for accurate font identification. We posit that this shortcoming stems from an insufficient

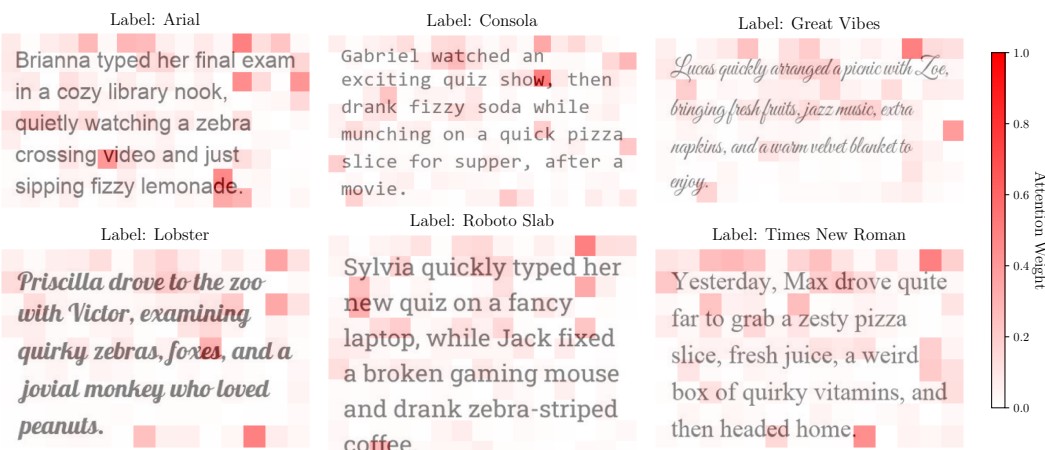

Figure 6: Attention heatmaps from the middle cross-attention layer ($14^{th}$) of QWEN2-VL-7B-INSTRUCT, illustrating how text tokens attend to image patches for six input images from the easy version of the benchmark, all of which the model fails to classify correctly. Attention weights are averaged across all heads.

alignment between visual attention mechanisms and the subtle typographic features that define font identity. Consequently, advancing font recognition within VLMs demands a more targeted modeling of local visual regions and a deeper integration of typographic priors.

# 8    Conclusion

In this paper, we introduce a specially designed Font Recognition Benchmark (FRB) comprising 15 popular font styles, designed with both easy and hard versions to comprehensively assess the capabilities of contemporary VLMs in this fine-grained recognition task. We conduct extensive experiments with a range of VLMs and observe that these models consistently struggle with font recognition, particularly on the hard version of the benchmark, where the stroop effect substantially reduces accuracy. This finding highlights that VLMs remain susceptible to misleading visual cues even when performing semantically grounded tasks. To further examine the capacity of VLMs in this setting, we evaluate various inference strategies, including CoT prompting, Multiple-Choice Question answering, and Few-Shot Learning. However, our results demonstrate that these methods offer only marginal improvements in overall accuracy. Additionally, we categorize the error types exhibited by VLMs into four distinct categories and analyze the attention patterns of two publicly available VLMs. Our analysis reveals that these models fail to adequately focus on the edge information of characters in the input images, which impairs their ability to correctly identify font styles and renders them vulnerable to variations in texture content.

# 9    Limitation

In this paper, we propose a specially designed benchmark comprising two task versions to enable a comprehensive evaluation of VLMs' performance on font recognition and analyze the factors contributing to their suboptimal performance. However, our study still has the following limitations: (i) Due to budget constraints, we are unable to conduct few-shot inference on more powerful models such as CLAUDE-3.5-SONNET and GPT-4O. (ii) As certain high-performing models are closed-source, we are unable to analyze their attention matrices, limiting our understanding of their decision-making processes. (iii) Our experiments rely on straightforward prompts for inference. Designing more sophisticated prompts, such as guiding models to focus on key regions in the input image, may enhance their performance on this task. We plan to explore these directions in future work.

## Acknowledgments

The work is partially supported by the NSF of the United States Grant CRII 2451683, an NVIDIA Academic Grants Program, University of California at Merced, and a UC Merced Faculty Research Award. The views and conclusions are those of the authors and should not reflect the official policy or position of the U.S. Government.

## Ethics Statement

Ethical considerations play a central role in this research. All models used in this study are either open-weight or widely adopted within the scientific community, ensuring transparency and reproducibility. The proposed benchmark and evaluation framework aim to assess the capabilities of current VLMs on font recognition without introducing or reinforcing harmful biases. No personally identifiable information or sensitive data is involved in this work. We are committed to responsible research practices, and we advocate for the transparent reporting and ethical deployment of AI technologies in ways that serve the broader interests of society.

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

# A  Font Introduction

Fonts are generally categorized into three major types: Serif Fonts, Sans-Serif Fonts, and Script & Decorative Fonts.

Serif fonts are well-suited for print materials, formal documents, and books, as their serifs help guide the reader's eye, improving the readability of long passages of text. Sans-serif fonts are commonly used in web design, presentations, and modern typographic layouts due to their enhanced clarity on digital screens. Script and decorative fonts are typically employed in posters, logos, and invitations, where individuality and artistic expression are prioritized.

We provide a brief introduction to all fonts used in this study. Additionally, Table 4 presents statistical data on several popular fonts analyzed in our experiments (Solomons, 2023).

| Font Name | Usage Percentage | Popular in Industry | Year Created |
|---|---|---|---|
| Helvetica | 25% | Graphic Design | 1957 |
| Times New Roman | 20% | Publishing | 1932 |
| Arial | 15% | Business | 1982 |
| Roboto | 10% | Web Design | 2011 |
| Georgia | 5% | Digital Media | 1993 |
| Calibri | 5% | Office Use | 2007 |

Table 4: Some popular fonts used in the paper, along with their typical usage statistics.

## A.1  Serif Fonts

**Baskerville SC:** A classic serif font with small caps styling, Baskerville SC exudes elegance and traditional sophistication. It is ideal for formal documents, book titles, and branding.

**Georgia:** Designed for readability on screens, Georgia is a warm and robust serif font with a classic touch. Its sturdy letterforms make it suitable for both print and digital use.

**Merriweather:** A modern serif font with a slightly condensed design, Merriweather offers a pleasant reading experience. It is often used for digital content, thanks to its balanced contrast and legibility.

**Times New Roman:** A widely recognized serif font, Times New Roman is synonymous with formal and academic writing. Its narrow proportions make it efficient for fitting text into limited spaces.

**Roboto Slab:** A geometric slab-serif font, Roboto Slab combines a modern feel with a sturdy, industrial aesthetic. It is commonly used for branding, web design, and contemporary editorial layouts.

## A.2  Sans-Serif Fonts

**Arial:** One of the most widely used sans-serif fonts, Arial is known for its clean and simple design. It is commonly used in digital and print media due to its high legibility.

**Calibri:** A default font in Microsoft Office, Calibri features a modern and rounded design. It provides a smooth reading experience and is often used in professional documents.

**Consola:** A monospaced sans-serif font, Consola is frequently used in coding environments. Its even spacing ensures clarity in programming and technical texts.

**Helvetica:** A timeless and versatile sans-serif font, Helvetica is known for its neutrality and balanced proportions. It is extensively used in corporate branding and signage.

**Monoton:** A decorative sans-serif font, Monoton features a retro, futuristic design with bold, high-contrast strokes. It is best suited for eye-catching headlines and artistic typography.

**Vera:** A clean and modern sans-serif font, Vera provides excellent readability and is commonly used in user interfaces. Its simplicity makes it adaptable to various design applications.

### A.3 Script & Decorative Fonts

**Great Vibes:** A flowing script font with elegant curves, Great Vibes is perfect for invitations and sophisticated branding. Its graceful letterforms add a touch of luxury to any design.

**Lobster:** A bold and stylish script font, Lobster features connected letterforms with a casual yet artistic feel. It is widely used in branding, menus, and creative projects.

**Pacifico:** Inspired by 1950s American surf culture, Pacifico is a playful script font with a relaxed, handwritten feel. It is often used for casual branding, logos, and fun designs.

**Satisfy:** A charming script font, Satisfy strikes a balance between casual and elegant lettering. It works well for branding, invitations, and creative typography.

## B Vision-Language Models

To effectively assess the font recognition capabilities of the latest VLMs, we conduct experiments on a selection of well-known models.

**Closed-Source Models:** We evaluate two multimodal models developed by OpenAI: GPT-4O and GPT-4O-MINI (OpenAI et al., 2024b; OpenAI, 2024), along with two widely recognized models from Anthropic and Google: CLAUDE-3.5-SONNET and GEMINI-2.0-FLASH-001 (Anthropic, 2024; Google, 2025).

**Open-Weight Models:** Our experiments include a diverse range of open-weight VLMs, spanning nine models across multiple series. Specifically, we test:

- Three models from the Qwen multimodal series: QWEN2-VL-7B-INSTRUCT, QWEN2.5-VL-7B-INSTRUCT and QWEN2-VL-72B-INSTRUCT (Wang et al., 2024; Bai et al., 2025).

- Two models from the latest Llama 3.2 Vision series: LLAMA-3.2-90B-VISION-INSTRUCT and LLAMA-3.2-11B-VISION-INSTRUCT (Meta AI, 2024).

- Two models from the Phi series developed by Microsoft: PHI-3.5-VISION-INSTRUCT and PHI-3-VISION-128K-INSTRUCT (Abdin et al., 2024).

- Two models from the Idefics series: IDEFICS3-8B-LLAMA3 and IDEFICS2-8B (Tronchon et al., 2024).

Here we provide a brief introduction to all Vision-Language Models utilized in this study.

### B.1 Closed-Source Models

**GPT-4O:** Developed by OpenAI, GPT-4O is a state-of-the-art multimodal model capable of processing text and images efficiently. It enhances vision-language understanding with improved reasoning, speed, and adaptability.

**GPT-4O-MINI:** A compact version of GPT-4O, GPT-4O-MINI is designed for faster inference while maintaining strong multimodal capabilities. It is optimized for applications requiring efficiency without sacrificing quality.

**CLAUDE-3.5-SONNET:** Developed by Anthropic, CLAUDE-3.5-SONNET is a highly capable vision-language model known for its alignment with human intent. It provides strong contextual understanding and excels in multimodal reasoning tasks.

**GEMINI-2.0-FLASH-001:** Created by Google, GEMINI-2.0-FLASH-001 is an optimized vision-language model focused on rapid image-text processing. It is designed for applications that demand fast and efficient multimodal comprehension.

## B.2 Open-Weight Models

**QWEN2-VL-7B-INSTRUCT:** Part of the Qwen multimodal series, QWEN2-VL-7B-INSTRUCT is an open-weight vision-language model by Alibaba. It provides a balance between efficiency and accuracy for multimodal reasoning tasks.

**QWEN2.5-VL-7B-INSTRUCT:** A refined version of its predecessor, QWEN2.5-VL-7B-INSTRUCT introduces improved visual understanding and stronger text-image alignment. It is designed for tasks requiring high accuracy in multimodal analysis.

**QWEN2-VL-72B-INSTRUCT:** The most powerful model in the Qwen2 series, QWEN2-VL-72B-INSTRUCT features a larger architecture for superior vision-language comprehension. It excels in complex tasks that require deep reasoning across modalities.

**LLAMA-3.2-90B-VISION-INSTRUCT:** A high-capacity model from Meta's Llama series, LLAMA-3.2-90B-VISION-INSTRUCT is designed for advanced vision-language interaction. Its large-scale architecture enhances performance in multimodal understanding.

**LLAMA-3.2-11B-VISION-INSTRUCT:** A smaller variant of the Llama 3.2 Vision series, LLAMA-3.2-11B-VISION-INSTRUCT offers a balance between efficiency and capability. It is ideal for resource-constrained environments requiring vision-language processing.

**PHI-3.5-VISION-INSTRUCT:** Developed by Microsoft, PHI-3.5-VISION-INSTRUCT is a lightweight yet powerful model tailored for multimodal tasks. It is optimized for efficiency while maintaining strong vision-language reasoning.

**PHI-3-VISION-128K-INSTRUCT:** A larger-scale version in the Phi series, PHI-3-VISION-128K-INSTRUCT supports longer context lengths, making it suitable for detailed and complex multimodal tasks. It enhances vision-language applications with broader contextual awareness.

**IDEFICS3-8B-LLAMA3:** A vision-language model from the Idefics series, IDEFICS3-8B-LLAMA3 focuses on open-ended multimodal reasoning. It is designed to handle diverse vision-language tasks with improved contextual understanding.

**IDEFICS2-8B:** Another model from the Idefics family, IDEFICS2-8B provides a strong foundation for vision-language applications. It is optimized for tasks requiring nuanced text-image comprehension.

## C Implementation Details

### C.1 Metric

We use the most straightforward metric: **Accuracy**. The model's performance is assessed by extracting the predicted labels from the text output and comparing them to the ground

truth labels using string matching. The accuracy of a model is calculated as follows:

$$\text{Accuracy} = \frac{1}{N} \sum_{i=1}^{N} \text{I}(y_i = \hat{y}_i) \tag{1}$$

where $\text{I}(\cdot)$ is an indicator function that is 1 if the condition is true and 0 otherwise, $y_i$ represents the true label, $\hat{y}_i$ is the predicted label, and $N$ is the total number of samples.

### C.2 Model Implementation

For all VLMs used in the experiments, we set the *temperature* to 0.0 and *top_p* to 0.95 to ensure faithfulness and minimize generation randomness, thereby enhancing reproducibility.

For open-weight models, we obtain the weights from HuggingFace and perform inference using *bfloat16* precision. All experiments involving these models are mainly conducted on a setup with four NVIDIA A100 80GB GPUs.

For closed-source models, we access the official API for inference. All experiments are conducted between January 25, 2025, and March 25, 2025. To compute the similarity between images for selecting few-shot demonstrations, we employ the **openai/clip-vit-base-patch32** model on HuggingFace.

## D Capability Gaps Between Open-weight and Closed-source Models

The experimental results indicate that contemporary VLMs struggle with font recognition tasks, with a significant accuracy gap persisting between open-weight and closed-source models.

In the sentence-based font recognition task, which is more commonly encountered in real-world scenarios, three out of four closed-source VLMs achieve at least 18% accuracy across various inference settings, with the best-performing model reaching nearly 67% under the multiple-choice question (MCQ) setting. In contrast, among all open-weight VLMs evaluated in this study, only LLAMA-3.2-11B-VISION-INSTRUCT and QWEN2-VL-72B-INSTRUCT attain approximately 18% accuracy, while the remaining models fall below 10%, with some approaching 0% accuracy. These findings highlight the substantial performance disparity between open-weight and closed-source VLMs in font recognition.

For the more challenging version of the task, overall performance remains low across both open-weight and closed-source models. The hard version introduces the stroop effect, adding an additional cognitive challenge that negatively impacts nearly all VLMs. Under the MCQ setting, all closed-source models are able to achieve accuracy above 6.67%, indicating that, in some cases, they successfully identify font styles rather than being misled by the textual content in the image. In contrast, open-weight models fail to do so in almost all cases, with the only exception of LLAMA-3.2-90B-VISION-INSTRUCT.

## E Comparative Analysis of Model Accuracy Across Different Fonts

To demonstrate the capability gap of VLMs in recognizing different font types, we evaluate 10 models across 15 distinct fonts and report the overall number of correct predictions under the four inference methods described in Table 1, on both the easy and hard versions of the benchmark.

From the results presented in Figure 11 to Figure 30, it is evident that each model tends to perform well on one or two specific font types, but struggles significantly with the remaining fonts. Notably, there is no single font or even a small subset of fonts that all models consistently recognize with high accuracy. Among the tested fonts, *Arial* and *Times New Roman* appear to be relatively easier for many models, likely due to their widespread use and higher representation in training corpora. In contrast, other fonts yield highly variable performance across models. For instance, LLAMA-3.2-90B-VISION-INSTRUCT correctly identifies 23 instances of the *Pacifico* font in the easy version of the benchmark, whereas

LLAMA-3.2-11B-VISION-INSTRUCT, despite belonging to the same model family, fails to recognize any instances under identical conditions. This highlights a critical limitation of current VLMs: they lack generalizable font recognition capabilities and exhibit substantial performance disparities across font types.

## F    Experimental Prompts

We present the simple prompts employed in the experiments shown in Figures 7 through 10.

> **Zero-Shot Prompt**
>
> What font is used in the text of the image?

Figure 7: Zero-Shot prompt for all experiments in this paper.

> **Zero-Shot CoT Prompt**
>
> What font is used in the text of the image? You need to think step by step.

Figure 8: Zero-Shot CoT prompt for all experiments in this paper.

> **Zero-Shot Multiple-Choice Questions Prompt**
>
> What font is used in the text of the image? You must choose one specific font name from the following list: [choices].

Figure 9: Zero-Shot Multiple-Choice Questions prompt for all experiments in this paper.

> **Zero-Shot CoT Multiple-Choice Questions Prompt**
>
> What font is used in the text of the image? You must choose one specific font name from the following list: [choices]. You need to think step by step.

Figure 10: Zero-Shot CoT Multiple-Choice Questions prompt for all experiments in this paper.

## G    Additional Attention Visualizations

Due to space constraints in the main part of the paper, we give additional visualizations for two open-weight models to illustrate their attention patterns on the same input image across different layers. For LLAMA-3.2-11B-VISION-INSTRUCT, we present attention matrices from all cross-attention layers, highlighting how attention evolves throughout the network. These visualizations reveal that, regardless of whether a layer is shallow or deep, the model consistently struggles to attend to the edge regions of characters in the image, which are critical for accurate font recognition.

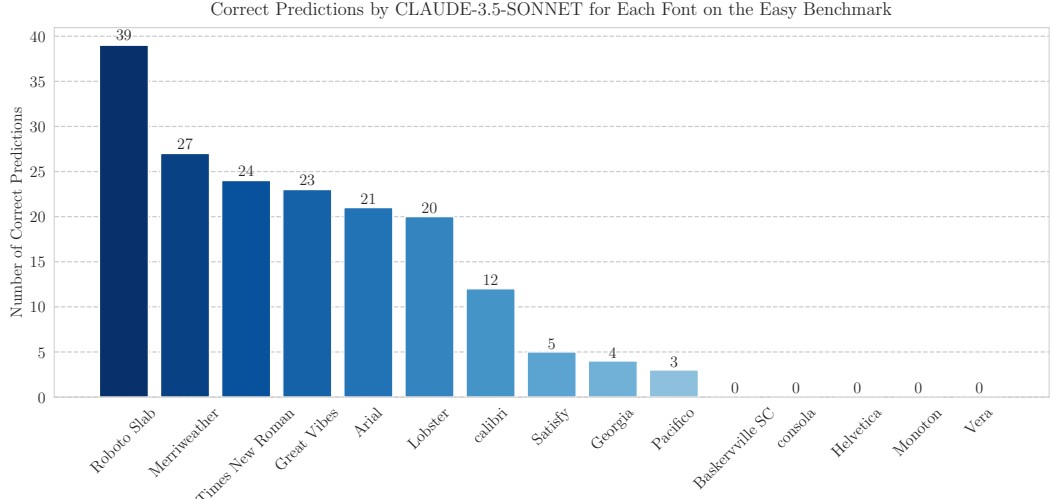

Figure 11: Number of correct predictions made by CLAUDE-3.5-SONNET on the Easy Benchmark for different fonts. The model performs best on Roboto Slab (39 correct predictions) and Merriweather (27 correct predictions), while struggling with fonts like Helevetica (0 correct predictions), Monoton (0 correct predictions), and Vera (0 correct predictions). The total number of experimental trials for each font is 40.

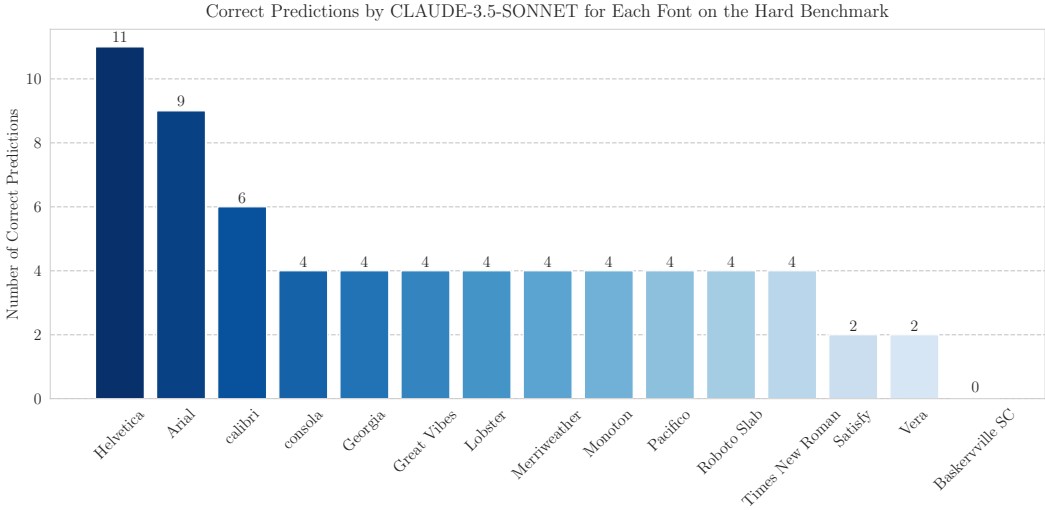

Figure 12: Number of correct predictions made by CLAUDE-3.5-SONNET on the Hard Benchmark for different fonts. The model performs best on Helvetica (11 correct predictions) and Arial (9 correct predictions), while struggling with fonts like Satisfy (2 correct predictions), Vera (2 correct predictions), and Baskerville SC (0 correct predictions). The total number of experimental trials for each font is 60.

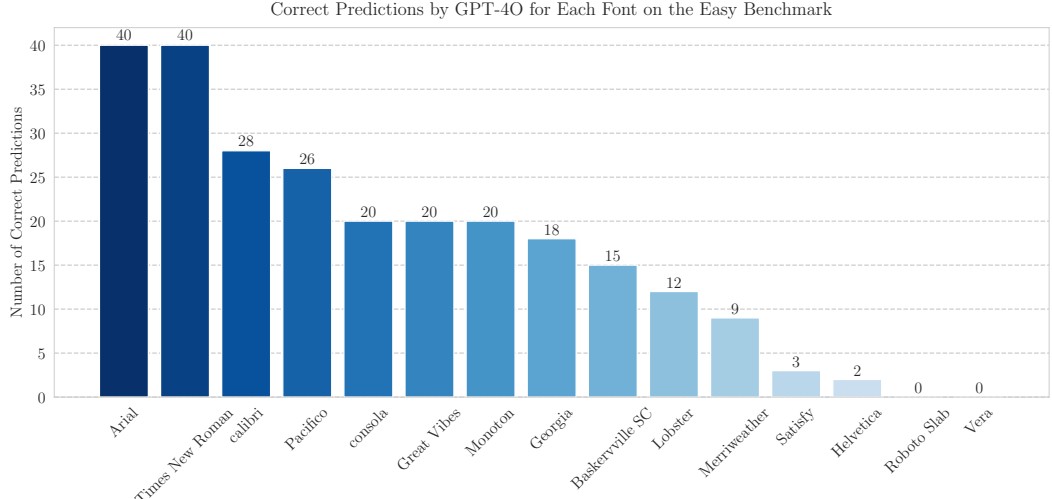

Figure 13: Number of correct predictions made by GPT-4O on the Easy Benchmark for different fonts. The model performs best on Arial (40 correct predictions) and Times New Roman (40 correct predictions), while struggling with fonts like Helevetica (2 correct predictions), Roboto Slab (0 correct predictions), and Vera (0 correct predictions). The total number of experimental trials for each font is 40.

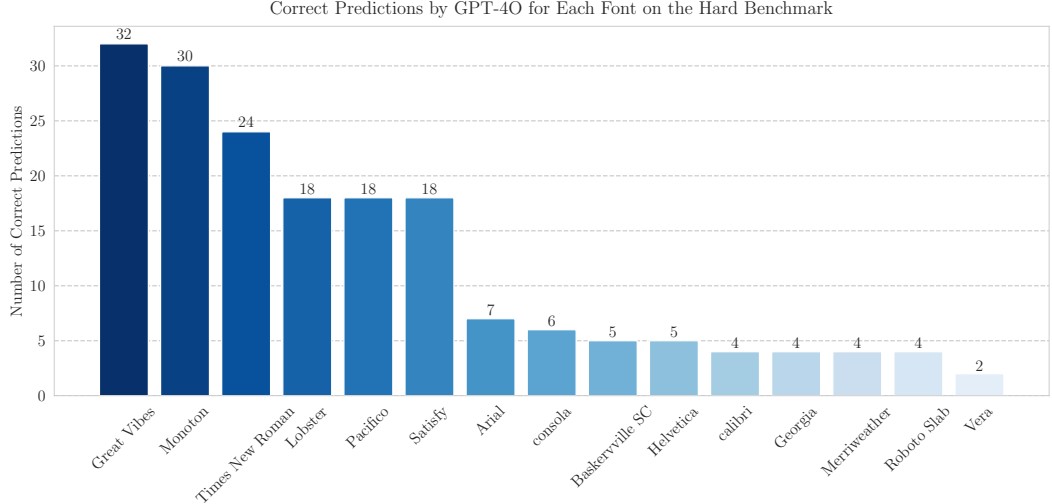

Figure 14: Number of correct predictions made by GPT-4O on the Hard Benchmark for different fonts. The model performs best on Great Vibes (32 correct predictions) and Monoton (30 correct predictions), while struggling with fonts like Merriweather (4 correct predictions), Roboto Slab (4 correct predictions), and Vera (2 correct predictions). The total number of experimental trials for each font is 60.

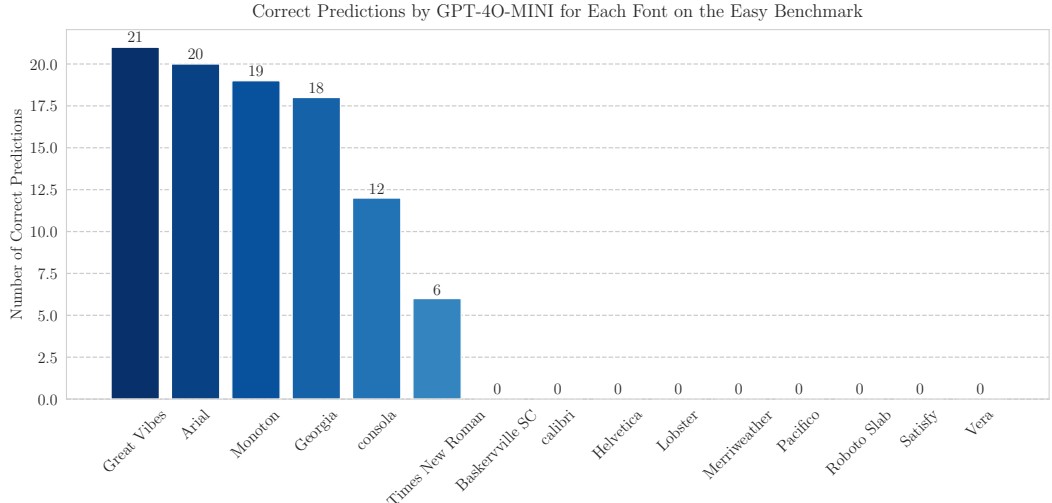

Figure 15: Number of correct predictions made by GPT-4O-MINI on the Easy Benchmark for different fonts. The model performs best on Great Vibes (21 correct predictions) and Arial (20 correct predictions), while struggling with fonts like Calibri (0 correct predictions), Lobster (0 correct predictions), and Pacifico (0 correct predictions). The total number of experimental trials for each font is 40.

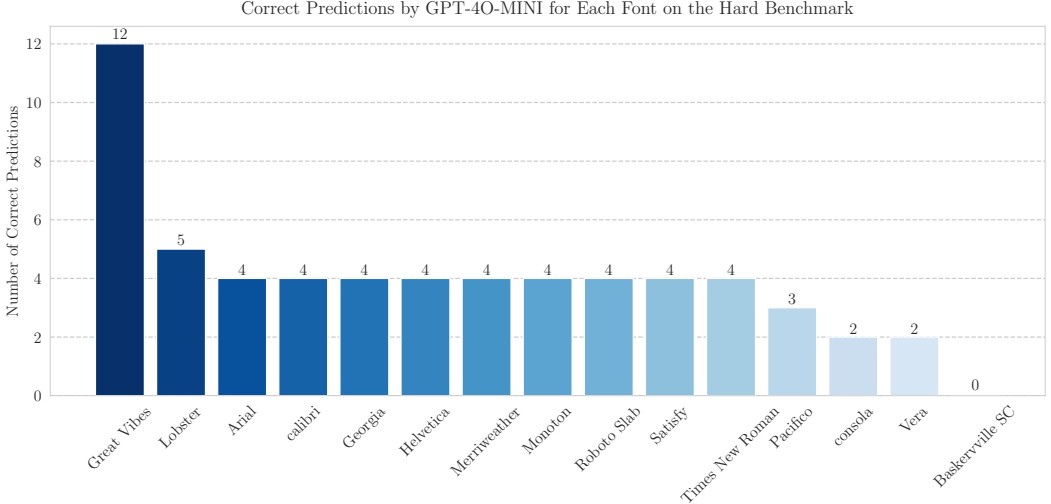

Figure 16: Number of correct predictions made by GPT-4O-MINI on the Hard Benchmark for different fonts. The model performs best on Great Vibes (12 correct predictions) and Lobster (5 correct predictions), while struggling with fonts like Consola (2 correct predictions), Vera (2 correct predictions), and Baskervville SC (0 correct predictions). The total number of experimental trials for each font is 60.

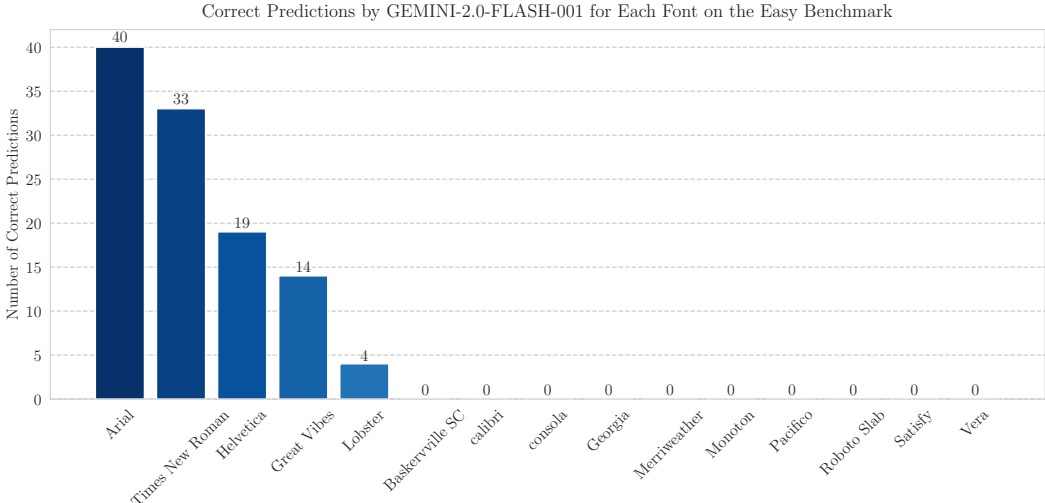

Figure 17: Number of correct predictions made by GEMINI-2.0-FLASH-001 on the Easy Benchmark for different fonts. The model performs best on Arial (40 correct predictions) and Times New Roman (33 correct predictions), while struggling with fonts like Calibri (0 correct predictions), Pacifico (0 correct predictions), and Satisfy (0 correct predictions). The total number of experimental trials for each font is 40.

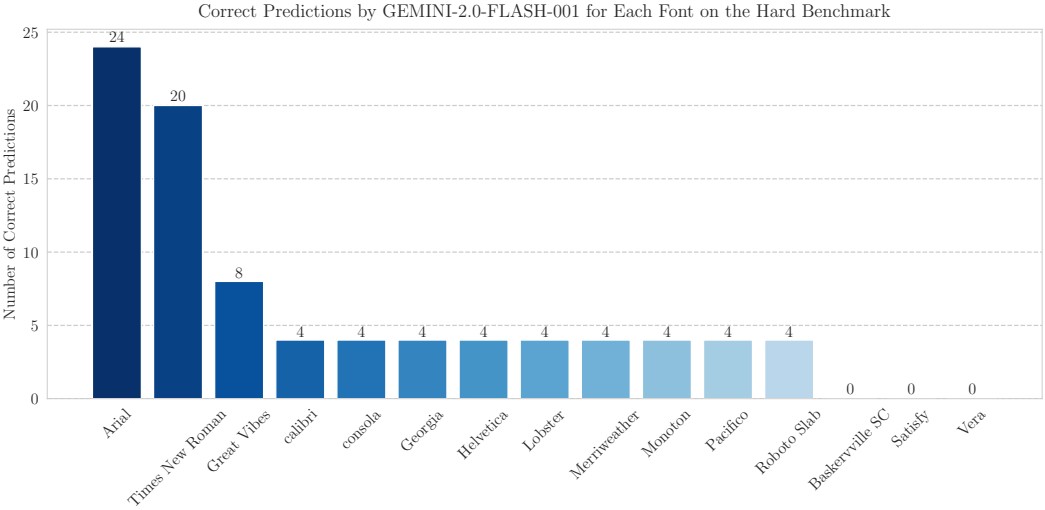

Figure 18: Number of correct predictions made by GEMINI-2.0-FLASH-001 on the Hard Benchmark for different fonts. The model performs best on Arial (24 correct predictions) and Times New Roman (20 correct predictions), while struggling with fonts like Baskervville SC (0 correct predictions), Satisfy (0 correct predictions), and Vera (0 correct predictions). The total number of experimental trials for each font is 60.

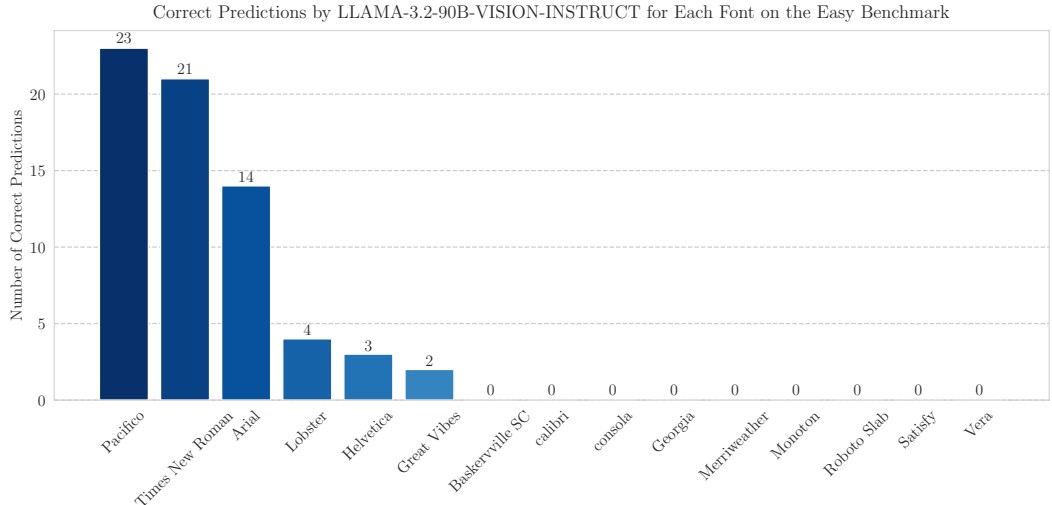

Figure 19: Number of correct predictions made by LLAMA-3.2-90B-VISION-INSTRUCT on the Easy Benchmark for different fonts. The model performs best on Pacifico (23 correct predictions) and Times New Roman (21 correct predictions), while struggling with fonts like Calibri (0 correct predictions), Consola (0 correct predictions), and Georgia (0 correct predictions). The total number of experimental trials for each font is 40.

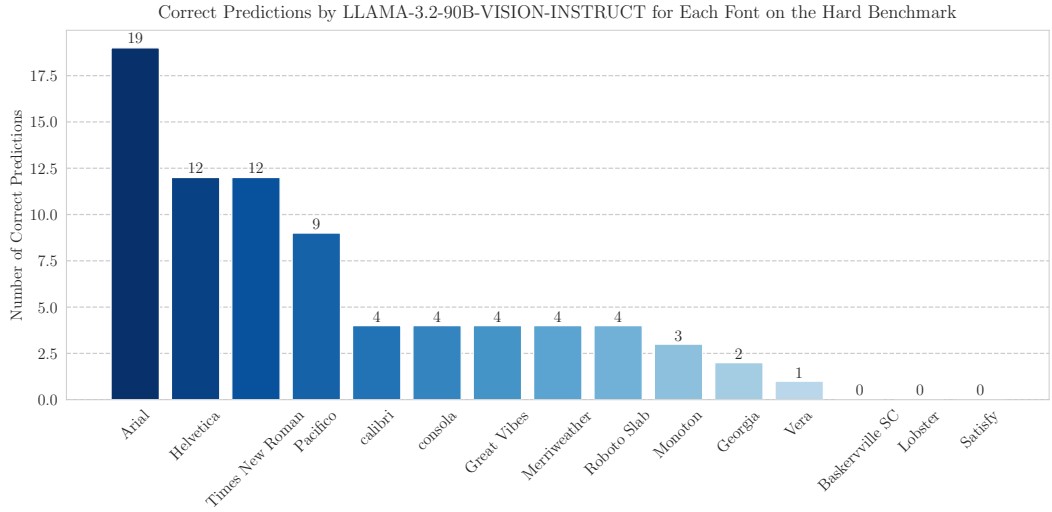

Figure 20: Number of correct predictions made by LLAMA-3.2-90B-VISION-INSTRUCT on the Hard Benchmark for different fonts. The model performs best on Arial (19 correct predictions) and Helvetica (12 correct predictions), while struggling with fonts like Baskervville SC (0 correct predictions), Satisfy (0 correct predictions), and Lobster (0 correct predictions). The total number of experimental trials for each font is 60.

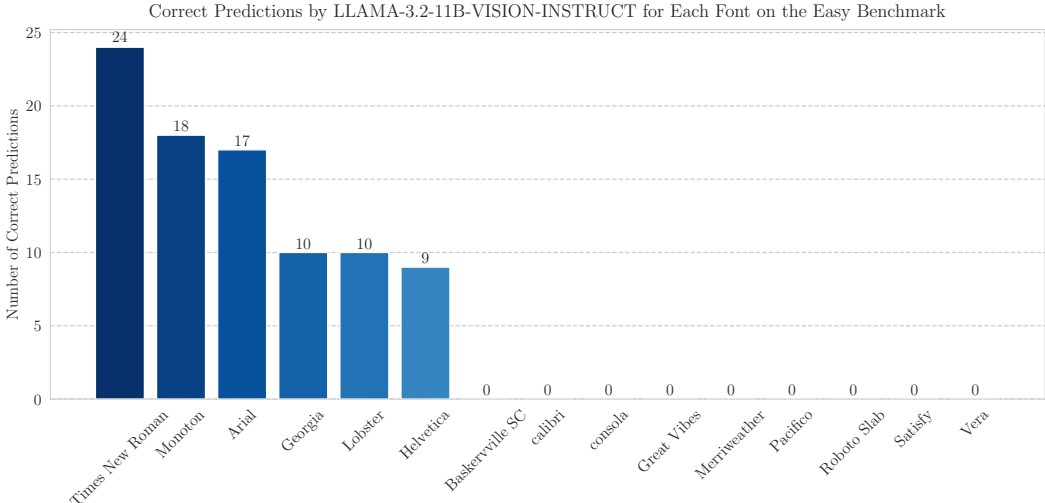

Figure 21: Number of correct predictions made by LLAMA-3.2-11B-VISION-INSTRUCT on the Easy Benchmark for different fonts. The model performs best on Times New Roman (24 correct predictions) and Monoton (18 correct predictions), while struggling with fonts like Calibri (0 correct predictions), Consola (0 correct predictions), and Vera (0 correct predictions). The total number of experimental trials for each font is 40.

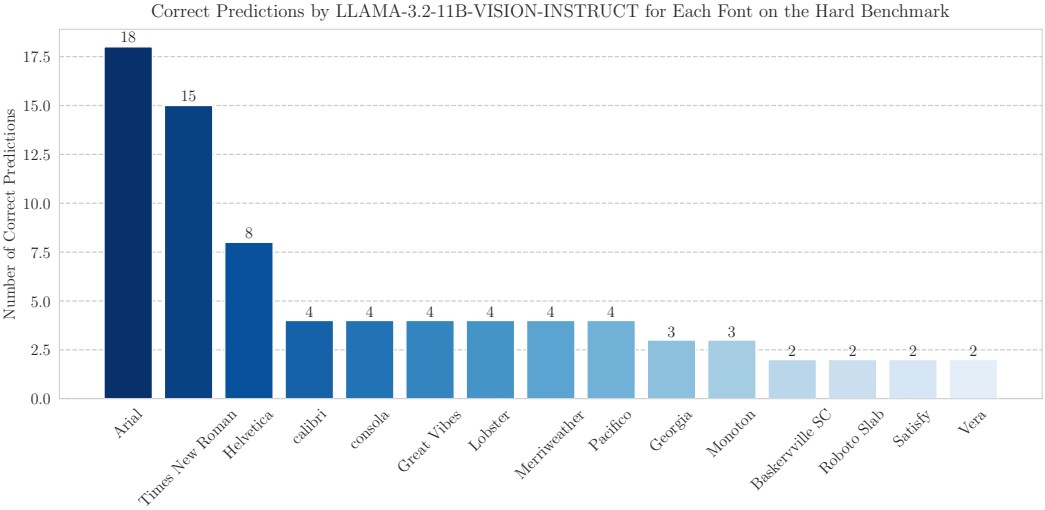

Figure 22: Number of correct predictions made by LLAMA-3.2-11B-VISION-INSTRUCT on the Hard Benchmark for different fonts. The model performs best on Arial (18 correct predictions) and Times New Roman (15 correct predictions), while struggling with fonts like Roboto Slab (2 correct predictions), Satisfy (2 correct predictions), and Vera (2 correct predictions). The total number of experimental trials for each font is 60.

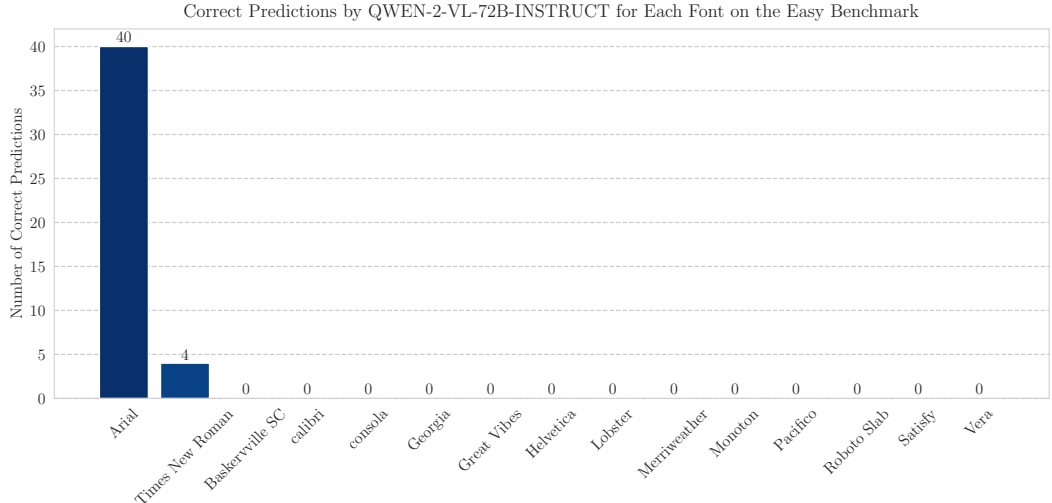

Figure 23: Number of correct predictions made by QWEN2-VL-72B-INSTRUCT on the Easy Benchmark for different fonts. The model performs best on Arial (40 correct predictions) and Times New Roman (4 correct predictions), while struggling with all remaining fonts. The total number of experimental trials for each font is 40.

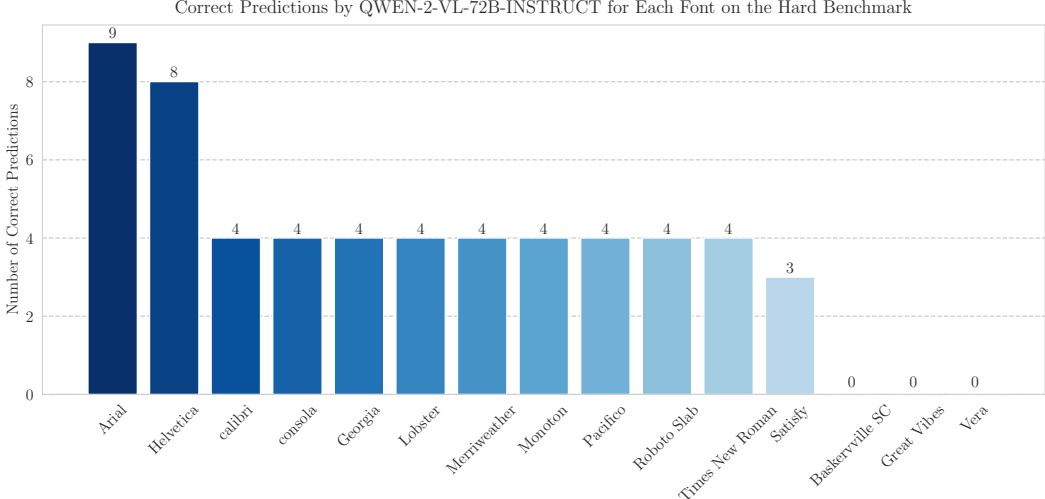

Figure 24: Number of correct predictions made by QWEN2-VL-72B-INSTRUCT on the Hard Benchmark for different fonts. The model performs best on Arial (9 correct predictions) and Helvetica (8 correct predictions), while struggling with fonts like Baskervville SC (0 correct predictions), Great Vibes (0 correct predictions), and Vera (0 correct predictions). The total number of experimental trials for each font is 60.

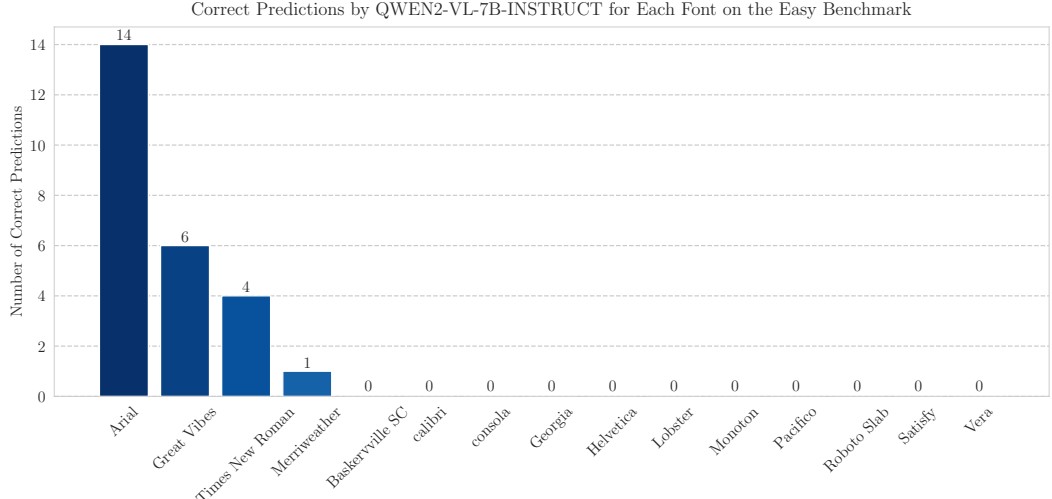

Figure 25: Number of correct predictions made by QWEN2-VL-7B-INSTRUCT on the Easy Benchmark for different fonts. The model performs best on Arial (14 correct predictions) and Great Vibes (6 correct predictions), while struggling with fonts like Calibri (0 correct predictions), Consola (0 correct predictions), and Georgia (0 correct predictions). The total number of experimental trials for each font is 40.

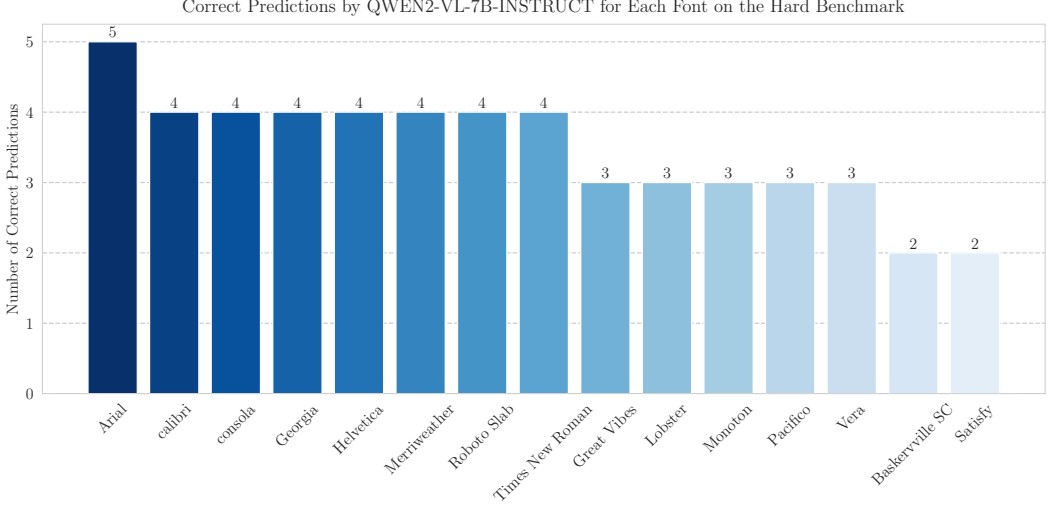

Figure 26: Number of correct predictions made by QWEN2-VL-7B-INSTRUCT on the Hard Benchmark for different fonts. The model performs best on Arial (5 correct predictions), while struggling with fonts like Baskervville SC (2 correct predictions), and Satisfy (2 correct predictions). The total number of experimental trials for each font is 60.

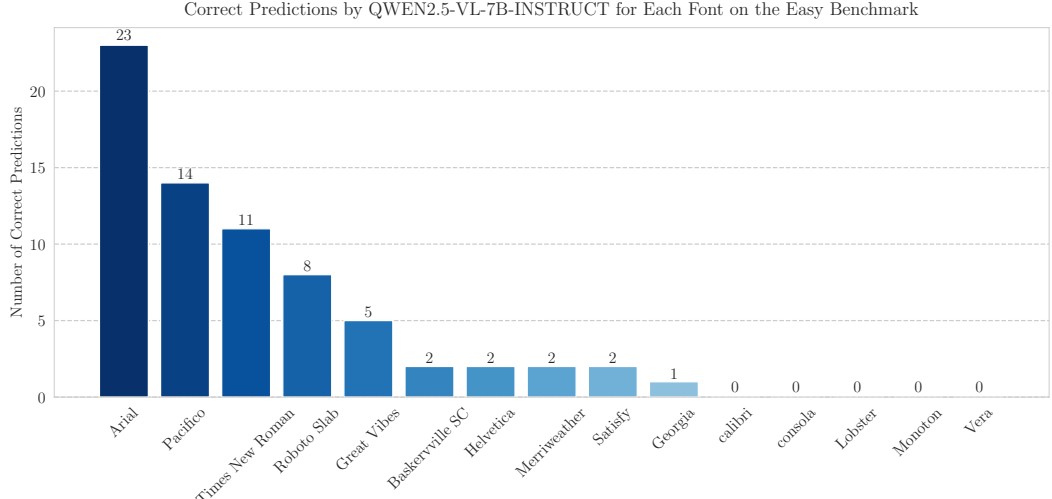

Figure 27: Number of correct predictions made by QWEN2.5-VL-7B-INSTRUCT on the Easy Benchmark for different fonts. The model performs best on Arial (23 correct predictions) and Pacifico (14 correct predictions), while struggling with fonts like Lobster (0 correct predictions), Monoton (0 correct predictions), and Vera (0 correct predictions). The total number of experimental trials for each font is 40.

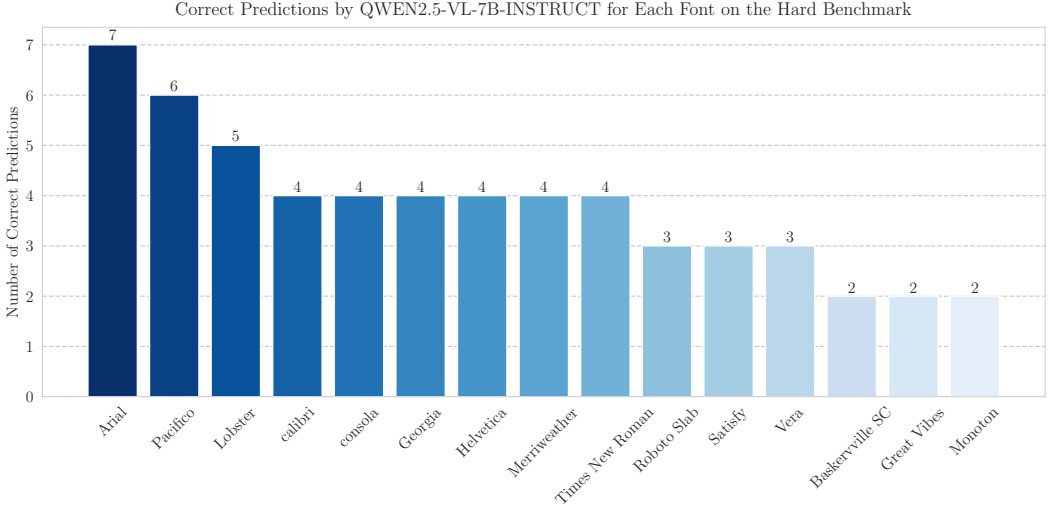

Figure 28: Number of correct predictions made by QWEN2.5-VL-7B-INSTRUCT on the Hard Benchmark for different fonts. The model performs best on Arial (7 correct predictions) and Pacifico (6 correct predictions), while struggling with fonts like Baskervville SC (2 correct predictions), Great Vibes (2 correct predictions), and Monoton (2 correct predictions). The total number of experimental trials for each font is 60.

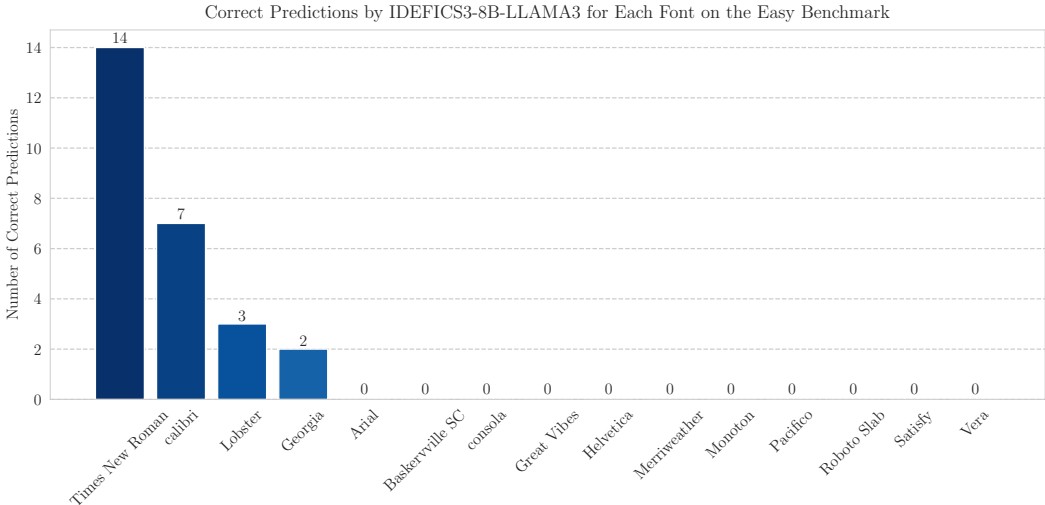

Figure 29: Number of correct predictions made by IDEFICS3-8B-LLAMA3 on the Easy Benchmark for different fonts. The model performs best on Times New Roman (14 correct predictions) and Calibri (7 correct predictions), while struggling with fonts like Consola (0 correct predictions), Monoton (0 correct predictions), and Vera (0 correct predictions). The total number of experimental trials for each font is 40.

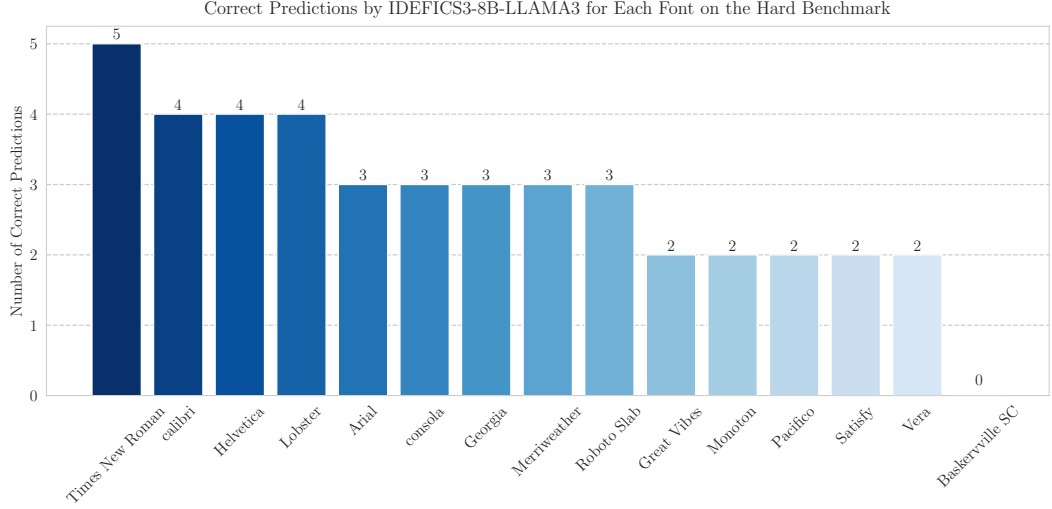

Figure 30: Number of correct predictions made by IDEFICS3-8B-LLAMA3 on the Hard Benchmark for different fonts. The model performs best on Times New Roman (6 correct predictions), while struggling with fonts like Baskervville SC (0 correct predictions). The total number of experimental trials for each font is 60.

Model: Llama-3.2-11B-Vision-Instruct && Ground Truth: Arial

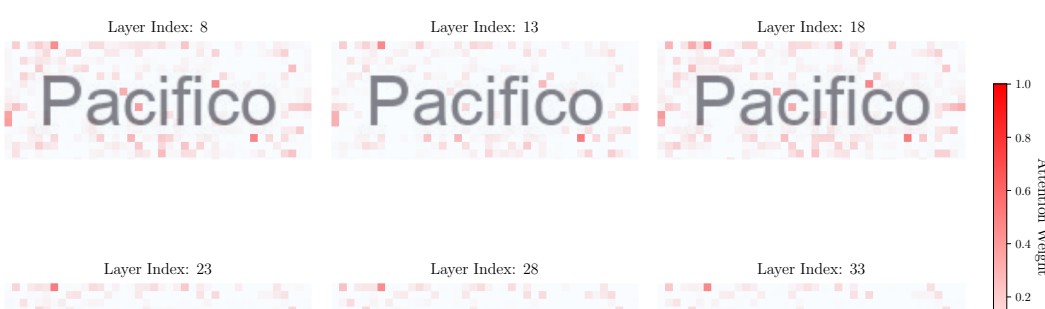

Figure 31: Visualization of attention weights from six cross-attention layers of LLAMA-3.2-11B-VISION-INSTRUCT (averaged across all attention heads), illustrating the progression of attention across the model's layers.

Model: Llama-3.2-11B-Vision-Instruct && Ground Truth: Baskervville SC

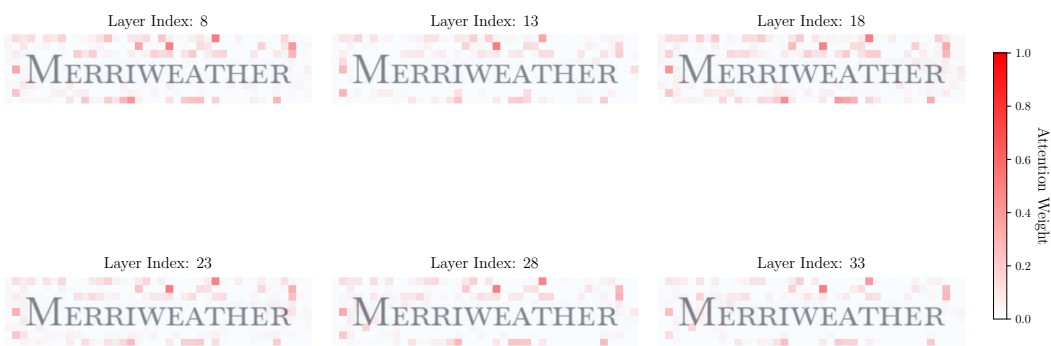

Figure 32: Visualization of attention weights from six cross-attention layers of LLAMA-3.2-11B-VISION-INSTRUCT (averaged across all attention heads), illustrating the progression of attention across the model's layers.

Model: Llama-3.2-11B-Vision-Instruct && Ground Truth: Roboto Slab

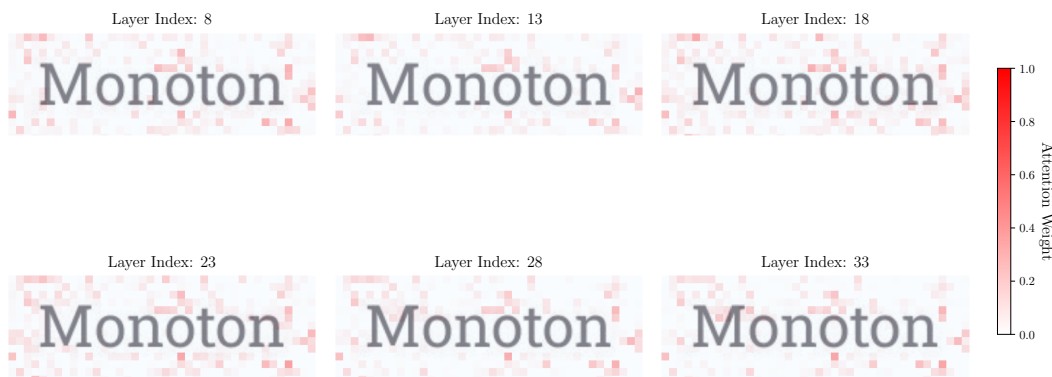

Figure 33: Visualization of attention weights from six cross-attention layers of LLAMA-3.2-11B-VISION-INSTRUCT (averaged across all attention heads), illustrating the progression of attention across the model's layers.

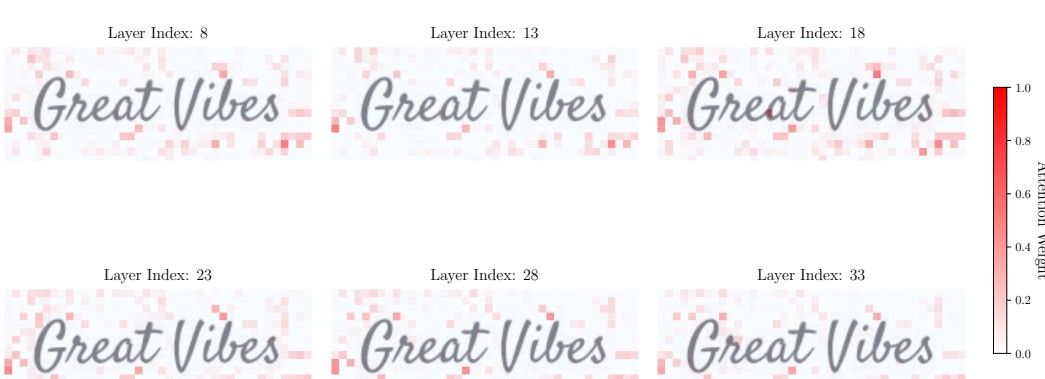

Figure 34: Visualization of attention weights from six cross-attention layers of LLAMA-3.2-11B-VISION-INSTRUCT (averaged across all attention heads), illustrating the progression of attention across the model's layers.

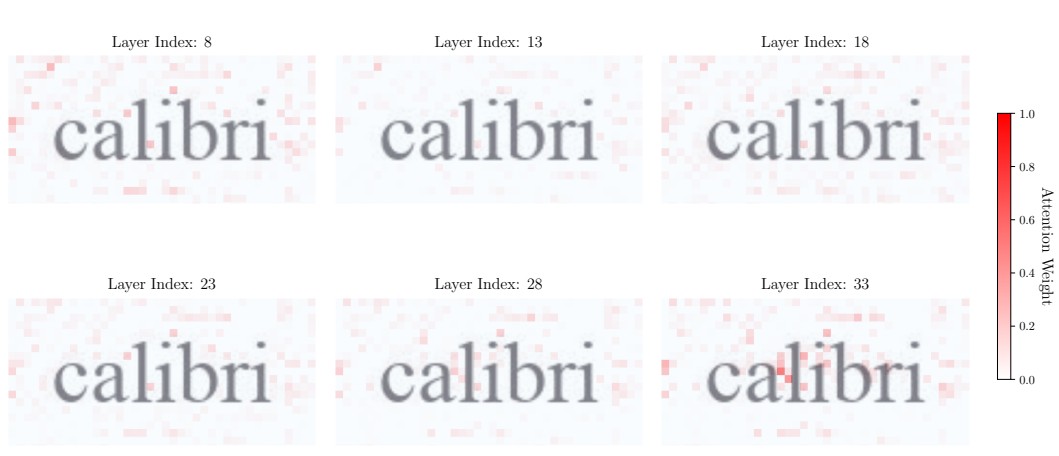

Figure 35: Visualization of attention weights from six cross-attention layers of LLAMA-3.2-11B-VISION-INSTRUCT (averaged across all attention heads), illustrating the progression of attention across the model's layers.

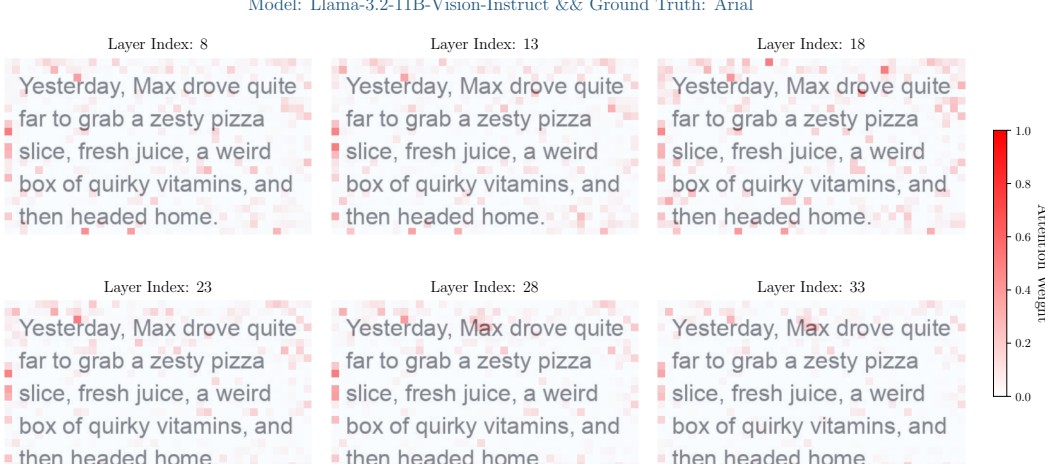

Figure 36: Visualization of attention weights from six cross-attention layers of LLAMA-3.2-11B-VISION-INSTRUCT (averaged across all attention heads), illustrating the progression of attention across the model's layers.

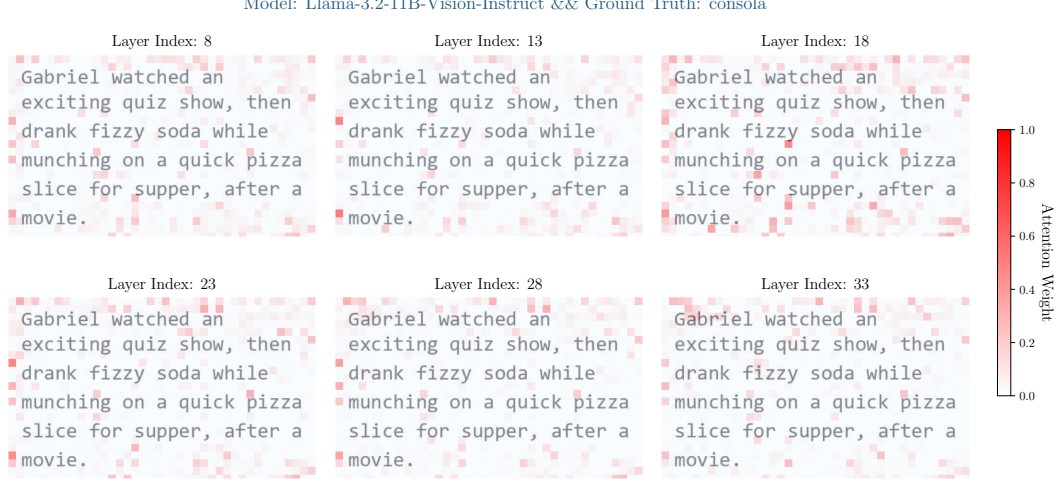

Figure 37: Visualization of attention weights from six cross-attention layers of LLAMA-3.2-11B-VISION-INSTRUCT (averaged across all attention heads), illustrating the progression of attention across the model's layers.

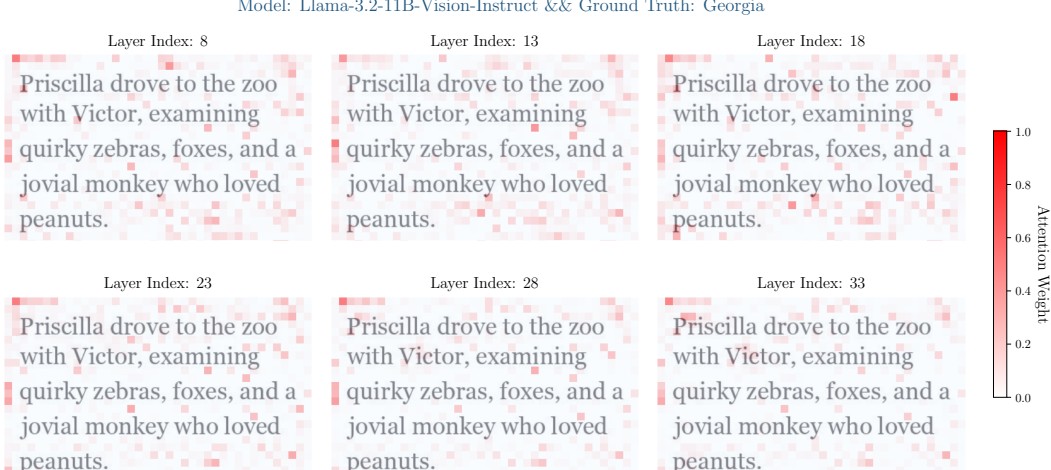

Figure 38: Visualization of attention weights from six cross-attention layers of LLAMA-3.2-11B-VISION-INSTRUCT (averaged across all attention heads), illustrating the progression of attention across the model's layers.

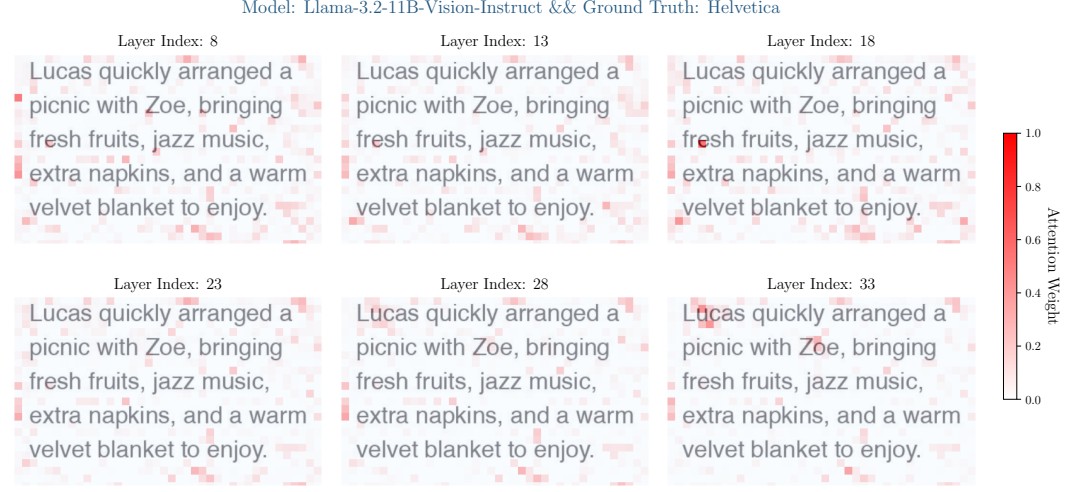

Figure 39: Visualization of attention weights from six cross-attention layers of LLAMA-3.2-11B-VISION-INSTRUCT (averaged across all attention heads), illustrating the progression of attention across the model's layers.

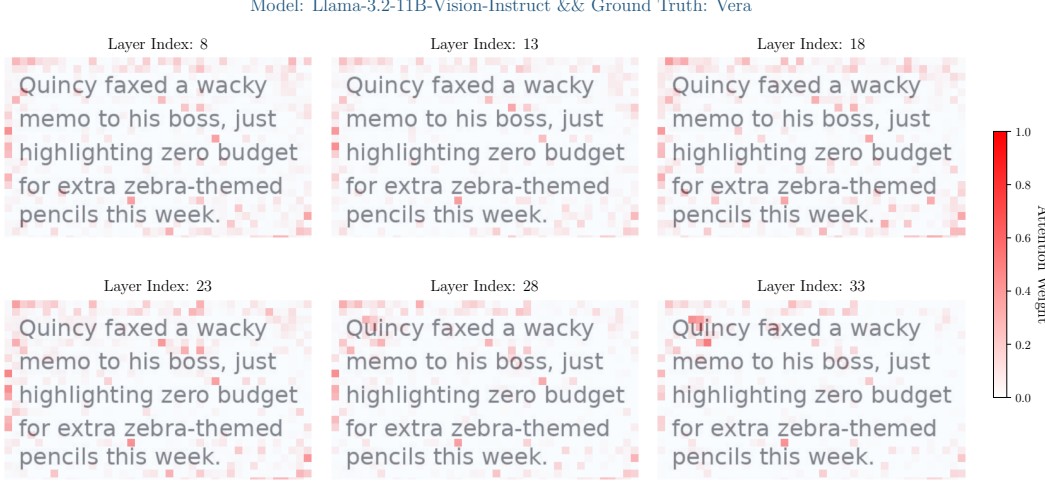

Figure 40: Visualization of attention weights from six cross-attention layers of LLAMA-3.2-11B-VISION-INSTRUCT (averaged across all attention heads), illustrating the progression of attention across the model's layers.

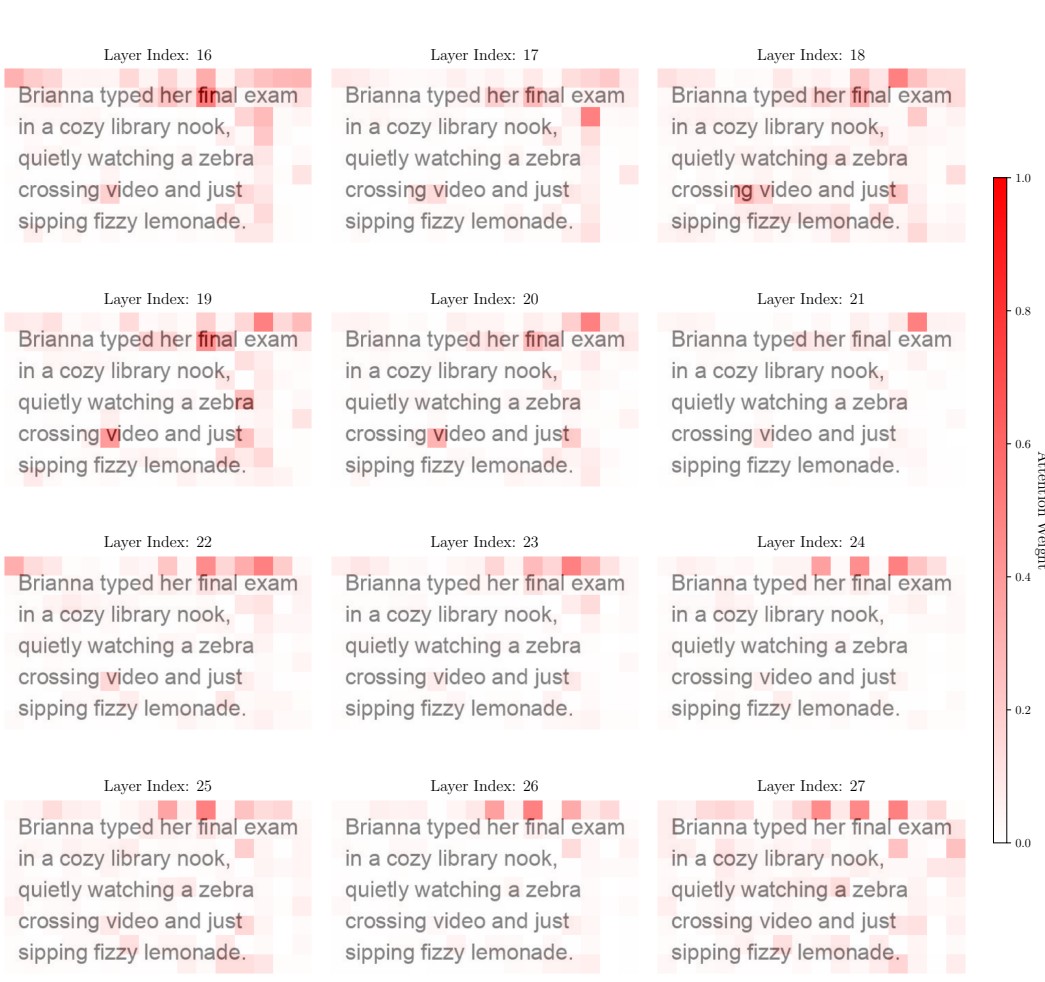

Figure 41: Visualization of attention weights from the 17th layer to the final attention layer of QWEN2-VL-7B-INSTRUCT (averaged across all attention heads), illustrating the progression of attention across the model's layers.

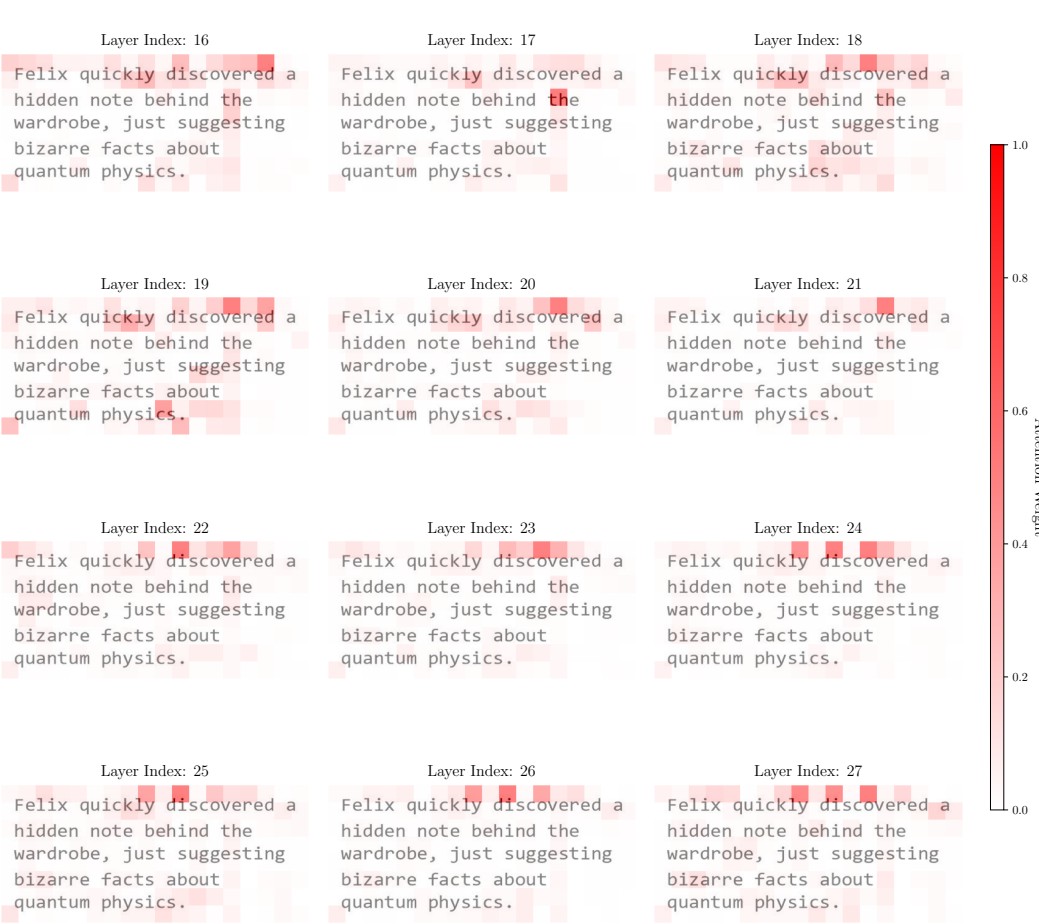

Figure 42: Visualization of attention weights from the 17th layer to the final attention layer of QWEN2-VL-7B-INSTRUCT (averaged across all attention heads), illustrating the progression of attention across the model's layers.

Figure 43: Visualization of attention weights from the 17th layer to the final attention layer of QWEN2-VL-7B-INSTRUCT (averaged across all attention heads), illustrating the progression of attention across the model's layers.

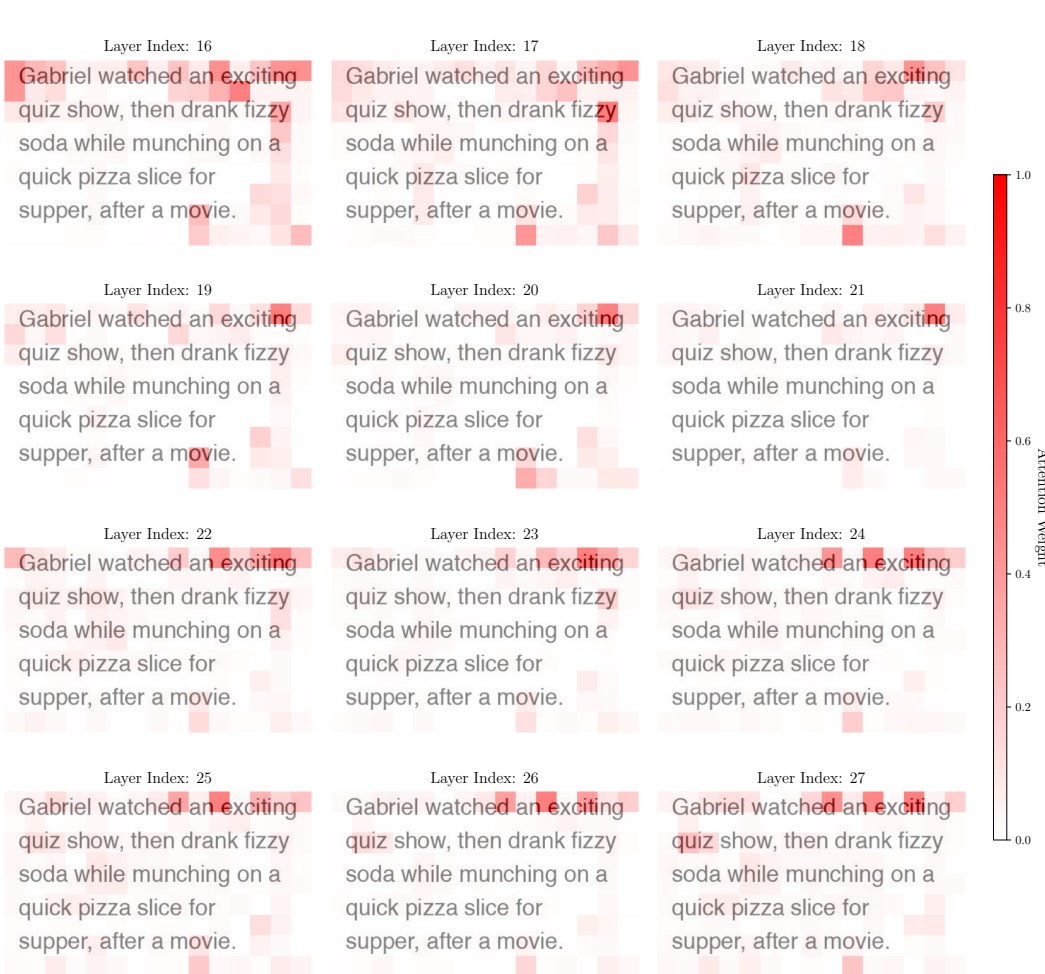

Figure 44: Visualization of attention weights from the 17th layer to the final attention layer of QWEN2-VL-7B-INSTRUCT (averaged across all attention heads), illustrating the progression of attention across the model's layers.

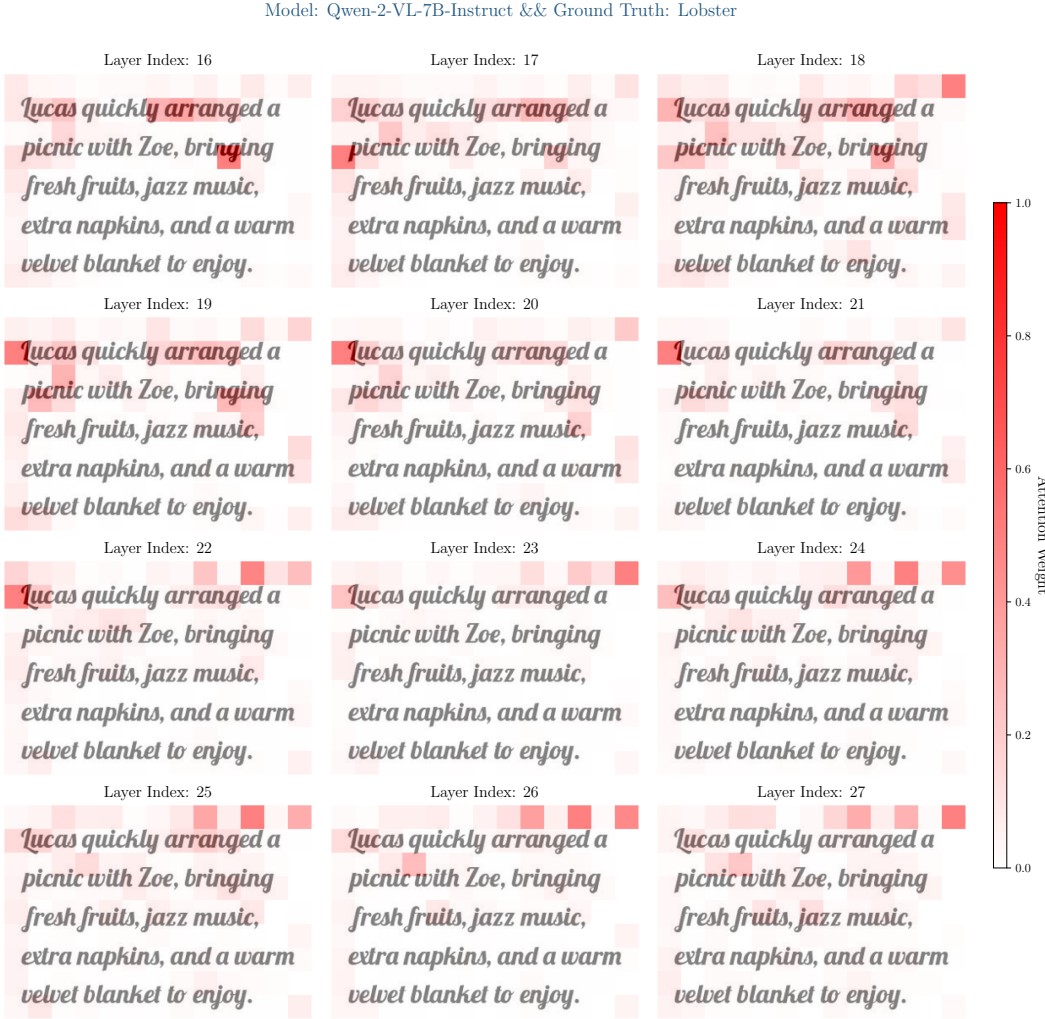

Figure 45: Visualization of attention weights from the 17th layer to the final attention layer of QWEN2-VL-7B-INSTRUCT (averaged across all attention heads), illustrating the progression of attention across the model's layers.

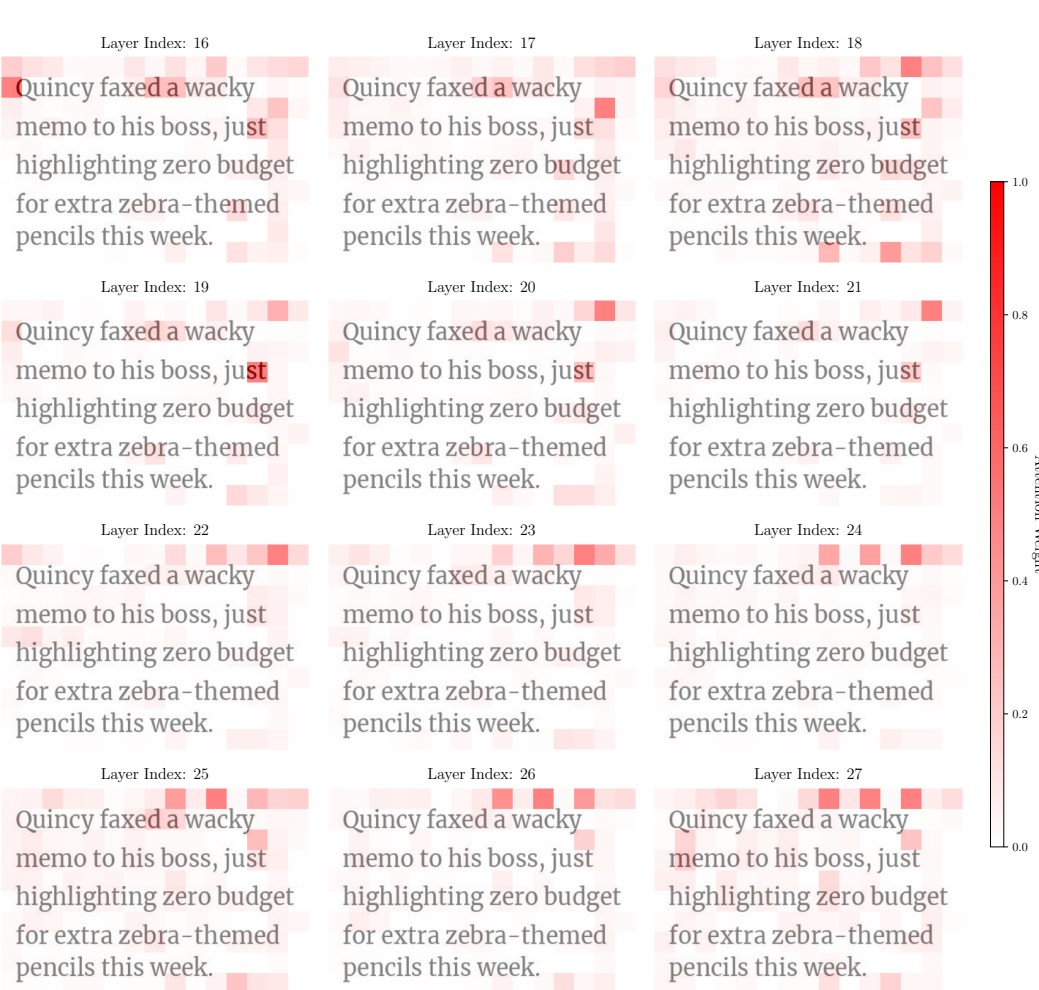

Figure 46: Visualization of attention weights from the 17th layer to the final attention layer of QWEN2-VL-7B-INSTRUCT (averaged across all attention heads), illustrating the progression of attention across the model's layers.

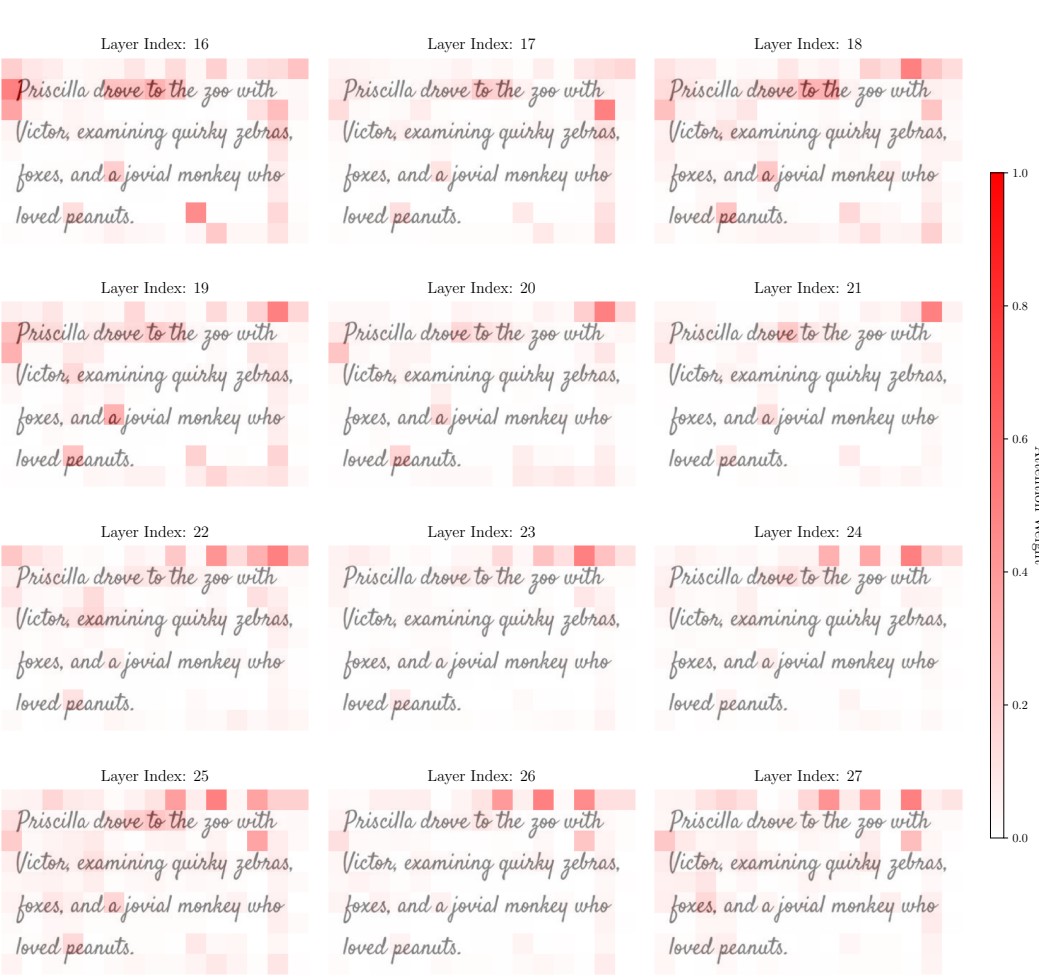

Figure 47: Visualization of attention weights from the 17th layer to the final attention layer of QWEN2-VL-7B-INSTRUCT (averaged across all attention heads), illustrating the progression of attention across the model's layers.

