# OpenReview forum: "Texture or Semantics? Vision-Language Models Get Lost in Font Recognition"
_colmweb.org/COLM/2025/Conference — COLM 2025_

### Official Review · Reviewer_A9ZJ · 2025-05-09

**Rating:** 8
**Confidence:** 5
**Ethics Flag:** 1

**Summary:**

The paper presents a new benchmark with 15 popular fonts, and shows the results from extensive benchmarking of visual language models in their ability to recognize the fonts. The practical need to recognize font is well motivated. The experiments are well executed and the paper is very clearly written. There is an effort to explain the poor performance of the models by analyzing their attention. This is the part I found least informative. The most interesting part is the hard condition, in which a font name is written in a different font. Results plummet in this condition.

**Questions To Authors:**

Is it really reasonable to expect that the ability to recognize fonts would emerge without any training on at somewhat related tasks? Is there a way to elicit the font names a model knows? I am not suggesting such experiments as condition for accepting the work, just wondering based on the presented content.

Can the CNNs mentioned in the intro be benchmarked against the VLLMs? Mostly I am looking for some sort of a strong take away or recommendation, like "include font-related task in the training mix", "it is not worth trying to coax VLLMs to do the task", or something of the sort that can serve researchers who may be interested in the task.

**Reasons To Accept:**

Well motivated, well executed experiments, described well.
The results are useful in documenting the weaknesses of current VLMs for the task.

**Reasons To Reject:**

Not really a reason to reject, but a drawback is that there is no proposal of how to teach models font recognition. A comparison with supervised simpler models is also missing.

---

> ### Author Response · Authors · 2025-06-01
>
> We thank Reviewer A9ZJ for your thoughtful and detailed feedback. We greatly appreciate your insights, which could help us improve the clarity and rigor of our study.
>
> **Q1:** It would be better to show the possibility of teaching VLMs font recognition.
>
> **R1:** Thanks for your suggestion! We add the experiment to demonstrate the possibility of using fine-tuning to teach VLMs in this field. We created 30 new sentences for 15 font types, in total 450 images for two VLMs, and we still do the test on the easy version of the benchmark. For fine-tuning, we use LoRA with rank 16 and alpha 16, we trained for two epochs with the cosine scheduler learning rate starting from 1e-4.
>
> The results are shown in the table below:
>
> | Model Name | zero-shot | zero-shot CoT | mcq zero-shot | mcq zero-shot CoT |
> |--|:--:|:--:|:--:|:--:|
> | Llama3.2-11B-Vision | 20.67 (+2.00)  | 9.33 (0.00) | 18.67 (+2.00) | 15.33 (+1.33) |
> | Qwen2.5-VL-7B | 13.33 (+1.33) | 6.67 (+1.34) | 16.67 (-0.66) | 11.33 (-0.67) |
>
> We could find that even after fine-tuning, the improvements are limited from VLMs, which demonstrates the drawback of VLMs in fine-grained tasks such as focusing on semantic cues. We would work on this as future work to explore effective ways of improving VLMs in this field.
>
> In summary, I would agree with your opinion of ‘include font-related tasks in the training mix’, we are interested in leaving this as future work.
>
> We follow your suggestion to add the experiment to improve our paper.
>
> **Q2:** It would be better to demonstrate that tested fonts are all in models’ knowledge scope..
>
> **R2:** Thanks for raising this consideration! The choice of the font used in the paper is based on the popularity rank given by the LLMs and some official websites, that’s to say, for all four closed-source models, they all know these 15 types of font and could give the right description of their features. However, their results on both easy and hard versions, especially even when we even give the multiple choices for them, are still poor.
>
> We follow your suggestion to add the discussion to improve our paper.
>
> **Q3:** It would be better to add the comparison with the CNN-based models.
>
> **R3:** Thanks for this helpful suggestion! We would like to give the accuracy of the CNN-based model. We created 100 new sentences for 15 font types, in total 1500 images for fine-tuning based on the HENet and ResNet backbone, which is designed for font recognition, the accuracy is only 39.33%, still not satisfied.
>
> Therefore, above two experiments demonstrate the issue of generalization limitation and training data consuming of the CNN-based model, which shows the necessity of considering well-generalized and strong capabilities VLMs to address these issues, also the first experiments post the question of how to effectively fine-tuning VLMs for this task, which we would work on as future work.
>
> We follow your suggestion to add the comparison to improve our paper.
>
> At last, we really thank the reviewer for recognizing the purpose and the significant meaning of our work on exploring the limitations of VLMs in fine-grained tasks like font recognition and their vulnerabilities towards the visual cues and semantic cues. Again, thanks for your valuable feedback and meaningful suggestions!

---

> > ### Comment · Reviewer_A9ZJ · 2025-06-03
> >
> > Thank you for presenting the additional experiments. Your work clearly identifies a weakness of current model. Hopefully follow up work will present compelling and practical solutions.  As in my original review, I believe this is a worthy paper to share with other researchers.

---

> > > ### Author Response · Authors · 2025-06-04
> > >
> > > We sincerely appreciate your recognition and acknowledgment of the contributions made in our paper. If possible, we kindly hope you may also consider increasing the rating accordingly, as this would be very helpful for the AC and SAC in conducting a more comprehensive evaluation.
> > >
> > > Thank you very much!

---

### Official Review · Reviewer_P7Qz · 2025-05-13

**Rating:** 5
**Confidence:** 3
**Ethics Flag:** 1

**Summary:**

This paper proposes two benchmarks to assess existing VLMs' ability in font recognition. The first is to predict one of 15 fonts from an image of a sentence, and the second is to predict a font from an image of a font. Since the word is about fonts, the second benchmark is more confusing. On these two benchmarks, this paper finds that existing VLMs perform worse and are easily deceived by the second benchmark. This paper further analyzes the categorization of errors and the attention map of cross-attention layers in VLMs.

**Questions To Authors:**

1. Which attention layer do you choose to print attention scores? How do you convert 2-dimensional matrices to 1-dimensional matrices?

2. What is the use of font recognition? Have other papers in this domain directly used or fine-tuned VLMs?

**Reasons To Accept:**

1. This paper identifies serious issues of VLMs' ability to recognize fonts.

2. This paper is easy to follow.

**Reasons To Reject:**

1. This paper does not compare previous work on font recognition, including benchmarks and specific models. Thus, I can't identify its unique and significant contribution. For example, is the previous benchmark suitable for evaluating LLMs?

2. This paper only compares features of CoT and few-shot and does not consider fine-tuning or test-time scaling. Fine-tuning is a reasonable way to perform specific tasks.

3. This paper exploits attention scores to explain why VLMs perform badly,  but even if the vlms perform good on one domain, their attention scores may not be distributed as you expect.

---

> ### Author Response · Authors · 2025-06-01
>
> We sincerely thank Reviewer P7Qz for your thoughtful and detailed feedback. We greatly appreciate your insights, which could help us improve the clarity and rigor of our study.
>
> **Q1:** It would be better to compare previous work on font recognition, including benchmarks and specific models.
>
> **R1:** Thanks for the consideration! The main purpose of our paper is to evaluate the capabilities of VLMs on fine-grained tasks like font recognition. Famous font recognition benchmarks like Adobe VFR and Explore All contain more than thousands of classes, which is not suitable for the evaluation of VLMs. We aim to use straightforward and simple methods, so our current benchmark only needs to focus on 15 popularly used fonts and with the design of stroop effect, already demonstrated the limitations of VLMs in this field.
>
> We would like to give the results of the CNN-based model. We created 100 new sentences for 15 font types, in total 1500 images for learning based on the HENet and ResNet backbone, but the accuracy is only 39.33%, still not satisfied. It shows the issue of generalization limitation and data consuming for the CNN-based model, which shows the necessity of considering well-generalized and strong capabilities VLMs to address these issues.
>
> **Q2:** It would be better to consider fine-tuning or reasoning models as possible solutions for improvement.
>
> **R2:** Our focus in this paper is to evaluate VLMs in zero-shot and prompt-based settings because these are the most common use cases for general-purpose models in practical deployments.We agree with the reviewer that considering above methods for comparison would further improve our paper.
>
> For reasoning models, we sample 2 images for each font from both easy and hard version, and the results are presented below:
>
> | Model Name | Easy Version | Hard Version |
> |--|:--:|:--:|
> | openai-o1-mini  | 23.33 | 6.67 |
> | openai-o3-mini | 26.67 | 10.00 |
> | gemini-2.5-pro | 16.67 | 10.00 |
>
> According to our experiments, reasoning models also do not show incredibly strong capabilities in the task and are still easy to be cheated by the texture information rather than semantic information.
>
> As for fine-tuning, we created 30 new sentences for 15 font types, in total 450 images for fine-tuning two VLMs. We use LoRA with rank 16 and alpha 16, trained for two epochs with the cosine scheduler learning rate starting from 1e-4.
>
> The results are shown in the table below:
>
> |Model Name|zero-shot|zero-shot CoT|mcq zero-shot|mcq zero-shot CoT|
> |--|:--:|:--:|:--:|:--:|
> |Llama3.2-11B-Vision| 20.67 (+2.00)  | 9.33 (0.00) | 18.67 (+2.00) | 15.33 (+1.33) |
> |Qwen2.5-VL-7B| 13.33 (+1.33) | 6.67 (+1.34) | 16.67 (-0.66) | 11.33 (-0.67) |
>
> We could find that even after fine-tuning, the improvements are limited from VLMs, which demonstrates the drawback of VLMs in fine-grained tasks such as focusing on semantic cues. We would work on this as future work to explore effective ways of improving VLMs in this field.
>
> **Q3:** It would be better to add discussion of using attention matrices for explanation.
>
> **R3:** We agree that attention maps do not always offer definitive explanations, and their interpretability can vary across models and tasks. However, our intent was not to treat attention as a perfect interpretive tool, but rather to use it as a proxy for identifying potential gaps in visual grounding.
>
> **Q4:** Which attention layer do you choose to print attention scores? How do you convert 2-dimensional matrices to 1-dimensional matrices?
>
> R4: For the Qwen model, which has 28 layers, since middle layers focus on visual cues, we calculate the average attention from layer index 7 to 20, and the attention score is the weights of the image patches to the “font” token. For the Llama model, we select layer index [13, 18, 23, 28], since the cross-attention layer in the Llama model appears every 5 layers from index 3 to 38.
>
> **Q5:** It’s better to add the discussion of choosing font recognition.
>
> **R5:** Font recognition presents key challenges in fine-grained visual perception, which is crucial for developing robust VLMs. Our work reveals VLMs' limitations in visual detail attention and their susceptibility to texture over semantics. Additionally, font recognition has important applications in design, OCR, accessibility, and content indexing, making it both theoretically and practically significant.
>
> **Q6:** It’s better to add more discussion about previous work.
>
> **R6:** To the best of our knowledge, there is limited to no prior work directly fine-tuning VLMs for font recognition. Existing approaches focused on CNN-based models trained specifically on labeled font datasets. Our work is among the first to systematically assess the capabilities of general-purpose VLMs in this fine-grained visual task and to highlight their vulnerability to semantic bias. This evaluation not only has practical meaning, but also exposes foundational limitations in their perceptual alignment and reasoning capabilities.

---

> > ### Comment · Reviewer_P7Qz · 2025-06-11
> > **Response to rebuttal**
> >
> > It would be better to quote my questions directly.
> >
> > As to R2,  what if you use more examples, e.g.,  million-level examples?
> >
> > Regarding R1 and R5,  have other papers discussed "Texture or Semantics" of VLMs on other tasks?
> >
> > As to R6,  why not use previous benchmarks?
> >
> > My reasons for rejection are similar to Reviewer FEXv. I agree that beyond these issues, this paper is well executed, it is clear, and the pointed issue is serious. I raised my score from 4 to 5.

---

### Official Review · Reviewer_FEXv · 2025-05-13

**Rating:** 6
**Confidence:** 3
**Ethics Flag:** 1

**Summary:**

This paper tests VLMs for their ability to recognise the font used in an image. For this purpose, the paper introduces a dataset that contains samples from a select set of fonts. As an additional twist, the paper also introduces a kind of stroop effect test, in that for each of the fonts there are images where the text set in this font is the name of a different font. The authors ran various experiments using frontier closed and open models, and testing single shot as well as few-shot and CoT prompting. The performance of even the biggest models is not great. An attention analysis shows that the models appear to focus on less relevant details.

**Questions To Authors:**

Please let us not normalise moving essential sections of the paper like the discussion of related work into the appendix. (Line 85.) The related work section that is in the main body of the paper is actually ok, so it's not a problem here, but let's just not start with this.

**Reasons To Accept:**

- clear question, well executed

**Reasons To Reject:**

- somewhat limited scope / special interest
- no comparison to specialised models
- discussion of wider impact (inability of model to reflect on the modality of information and resolve conflict) somewhat limited

---

> ### Author Response · Authors · 2025-06-01
>
> We thank Reviewer FEXv for your thoughtful and detailed feedback. We greatly appreciate your insights, which could help us improve the clarity and rigor of our study.
>
> **Q1:** It’s better to add the discussion for research scope.
>
> **R1:** We want to demonstrate that font recognition embodies fundamental challenges in fine-grained visual perception, which is an essential part for the development of robust, general-purpose VLMs. Our work aims to highlight limitations in VLMs’ attention to visual details and easily to be affected by the texture content rather than focusing on semantics. Moreover, font recognition has concrete applications in design, OCR, accessibility tools, and digital content indexing, making it both theoretically informative and practically relevant.
>
> We follow your suggestion to add the discussion to improve our paper.
>
> **Q2:** It would be better to add the comparison with specialized models.
>
> **R2:** Thanks for the suggestion! Our primary focus is to evaluate general-purpose VLMs to assess their suitability for a task that is both visually fine-grained and context-sensitive. However, we would like to add the following experiments!
>
> First, we fine-tune two VLMs to show whether specialized VLMs could give significant improvements. We created 30 new sentences for 15 font types, in total 450 images, and we still do the test on the easy version of the benchmark. For fine-tuning, we use LoRA with rank 16 and alpha 16, we trained for two epochs with the cosine scheduler learning rate starting from 1e-4.
>
> The results are shown in the table below:
>
> |Model Name|zero-shot|zero-shot CoT|mcq zero-shot|mcq zero-shot CoT|
> |--|:--:|:--:|:--:|:--:|
> |Llama3.2-11B-Vision| 20.67 (+2.00)  | 9.33 (0.00) | 18.67 (+2.00) | 15.33 (+1.33) |
> |Qwen2.5-VL-7B| 13.33 (+1.33) | 6.67 (+1.34) | 16.67 (-0.66) | 11.33 (-0.67) |
>
> We could find that even after fine-tuning, the improvements are limited from VLMs, which demonstrates the drawback of VLMs in fine-grained tasks such as focusing on semantic cues.
>
> Secondly, we would like to give the accuracy of the CNN-based model. We created 100 new sentences for 15 font types, in total 1500 images for learning based on the HENet and ResNet backbone, which is designed for font recognition, the accuracy is only 39.33%, still not satisfied.
>
> Above two experiments demonstrate the issue of generalization limitation and training data consuming of the CNN-based model, which shows the necessity of considering well-generalized and strong capabilities VLMs to address these issues, also the first experiments post the question of how to effectively fine-tuning VLMs for this task, which we would work on as future work.
>
> We follow your suggestion to add the comparison to improve our paper.
>
> **Q3:** It would be better to add discussion of modality conflict and reasoning limitations.
>
> **R3:** Thanks for this suggestion! We agree that the stroop effect in our paper offers a unique lens to examine how VLMs handle conflicting signals between visual features and textual semantics. We expand the discussion on this point, specifically elaborating on how the inability to suppress textual bias in the presence of conflicting visual cues reveals a gap in modality-aware reasoning. We frame this failure not just as a task-specific issue, but as a broader limitation in current multimodal architectures’ interpretability and alignment mechanisms.
>
> We follow your suggestion to add the discussion to improve our paper.
>
> At last, sorry for the inconvenience caused by moving the full related work part into appendix, we follow your suggestion to move this essential part back to the main body.
>
> Again, thanks for your valuable feedback and meaningful suggestions!

---

> > ### Comment · Reviewer_FEXv · 2025-06-05
> > **read response**
> >
> > Thanks for your response. The additional experiments will certainly put weight to your point.
> >
> > The claim that font recognition is an exemplary stand-in for various types of fine-grained visual recognition is interesting; but if you'd really like to make something out of it, you would need to show that this is the case by correlating performance with some other task. Without that, I would still maintain that font recognition isn't really a "fundamental task" -- it's a very specialised task that has some relevance for a well defined group of users.

---

> > > ### Author Response · Authors · 2025-06-07
> > >
> > > We sincerely thank you for your continued thoughtful engagement with our work. We appreciate your suggestion to include experiments on more fundamental tasks that may be influenced by fine-grained font features.
> > >
> > > Here, we present the experimental results on the OCR task. We generated 10 long passages using 10 selected fonts that differ significantly in style (with all image parameters kept constant) and used four closed-source models to perform OCR. The table below shows the average number of incorrect words produced during OCR for each font:
> > >
> > > | Font Name | GPT-4o | Claude-3.5-Sonnet | Gemini-2.0-Flash-001 | GPT-4o-mini |
> > > |--------------------|:-----:|:------:|:------:|:-----:|
> > > | Arial                |   1.7   |   2.3    |   1.9    |   2.1   |
> > > | Baskerville SC       |   2.0   |   1.6   |   2.0   |  1.9  |
> > > | calibri              |   1.5   |   1.9   |   1.7   |   1.6  |
> > > | Times New Roman      |   1.3   |   1.0   |   1.2   |  1.2  |
> > > | Vera                 |   1.5   |   1.2   |  1.4  |  1.8   |
> > > | Great Vibes          |   3.1   |   3.9    |   2.8   |  3.5  |
> > > | Lobster              |   2.7   |   2.6    |   3.1   |  2.9  |
> > > | Monoton              |  4.0   |   4.5   |  3.6   |  4.1  |
> > > | Pacifico             |   2.8   |   3.1   |   2.3   |   3.4  |
> > > | Satisfy              |  2.3   |   2.0    |   2.1   |   2.6   |
> > >
> > > From the results above, we can observe a clear performance gap in OCR accuracy between Serif or Sans-Serif fonts and Script fonts. The five Script fonts listed below produce significantly more errors, even when applied to the same passages. Among them, Monoton performs the worst due to its highly distinctive style. This experiment demonstrates that font characteristics can actually influence the performance of OCR tasks.
> > >
> > > We have followed your suggestion and added this experiment to our paper to improve it. Thanks for your valuable suggestion!

---

### Official Review · Reviewer_A1JS · 2025-05-15

**Rating:** 5
**Confidence:** 4
**Ethics Flag:** 1

**Summary:**

This paper introduces a font recognition benchmark comprising 15 common fonts to evaluate the font recognition capabilities of 13 VLMs. The dataset is divided into two difficulty levels: in the easy version, sentences are rendered in the specified font; in the hard version, the font names are rendered in different font. Experimental results demonstrate that current VLMs perform poorly on this task, and that additional techniques such as chain-of-thought prompting or few-shot learning do not yield substantial improvements. VLM errors in font recognition are categorized into four types. Attention matrices of two open-source VLMs are analyzed.

**Reasons To Accept:**

1) This paper systematically investigates VLMs’ performance on font recognition by proposing a benchmark with two levels of difficulty, exploring the Stroop Effect’s impact on model behavior.
2)This paper evaluates 13 mainstream VLMs, including both open-source and closed-source models, offering a thorough performance comparison.
3) This paper provides visualizations of the attention matrices for LLAMA-3.2-11B-VISION-INSTRUCT and QWEN2-VL-7B-INSTRUCT, shedding light on how these models attend to font cues.

**Reasons To Reject:**

1) This paper requires VLMs to precisely distinguish certain fonts, which may be overly demanding due to visually near-identical typefaces (e.g., Helvetica vs. Arial). For the “Indecisive Classification” error category, it would be informative to know whether the incorrect predictions are visually similar to the target font.
2) The paper lacks attention matrices visualizations for QWEN2-VL-7B-INSTRUCT on the hard version of the task, leaving that part of the study incomplete.
3) The benchmark could be more comprehensive. The easy version uses full sentences, while the hard version uses single words. Sentence format can also be used to introduce the Stroop Effect. Additionally, font size is not controlled across the two versions, potentially confounding the results.

---

> ### Author Response · Authors · 2025-06-01
>
> We thank Reviewer A1JS for your thoughtful and detailed feedback. We greatly appreciate your insights, which could help us improve the clarity and rigor of our study.
>
> **Q1:** It would be better to further discuss the necessity of difficulty for near-identical typefaces.
>
> **R1:** Thanks for this consideration! We agree that fine-grained distinctions can be challenging due to their visual similarity. However, this difficulty is precisely what motivates our investigation. One of our aims is to assess whether modern VLMs, which claim strong visual abilities, can handle such realistic and nuanced challenges. Also, the easy version of the task includes numerous letters to let VLMs observe the edge parts, suggesting that if VLMs genuinely possess the ability, differentiating them should not be challenging, which is also the reason why we designed two versions of the dataset. Additionally, the limited increase brought by few-shot learning also demonstrates the inability of VLMs on learning from the semantic cues. In summary, our design of experiments and benchmarks points out the limitations of VLMs in figuring out semantic cues and their vulnerabilities towards texture cues.
>
> We follow your suggestion to add the discussion to improve our paper.
>
> **Q2:** It would be better to further analyze “Indecisive Classification” errors.
>
> **R2:** Thanks for the suggestion! We agree that this could help make our paper more comprehensive in analyzing VLMs’ weakness in font recognition. Experimental results show that in the ‘Indecisive Classification’ category, only 23% of the response appears to have a font prediction, the rest responses are only able to point out the ‘large’ category such as Serif or Sans-Serif. That’s to say, in most situations the models are even difficult to give related candidates.
>
> We follow your suggestion to add the analysis to improve our paper.
>
> **Q3:** It would be better to add visualizations for QWEN2-VL-7B-INSTRUCT on the hard version.
>
> **R3:** Thanks for the helpful suggestion! We have already plotted all the attention figures for the false cases, the observations and conclusions are the same as the figures shown in the paper right now. Considering the length of the paper, we didn’t add them into the paper, but we would follow your suggestion to add them into the appendix to improve our paper!
>
> **Q4:** It would be better to additionally introduce the stroop effect in the sentence format.
>
> **R4:** Thanks for this suggestion! We currently choose the word-level stroop effect since it is straightforward and we increase the font size for this hard version to expect that model could possibly learn from the edge parts but the results are disappointing. We already added another test dataset that incorporates stroop effect into sentence format.
>
> We replace the font name with the sentence introduction of the font, such as: “We introduce Times New Roman, which is a type of font that …”, in total it’s still 225 images. We keep the font size as the easy version uses. And we present the results in the below table:
>
> | Model Name | zero-shot | zero-shot CoT | mcq zero-shot | mcq zero-shot CoT |
> |----------|:---------:|:-------------:|:-------------:|:------------------:|
> | GPT-4o      | 12.89      | 14.22          | 22.22          | 24.44              |
> | Claude-3.5-Sonnet   | 8.89      | 9.78          | 13.78          | 13.33              |
> | Gemini-2.0-Flash-001 | 9.78      | 8.89          | 11.11          | 9.78            |
> | GPT-4o-mini    | 7.11      | 8.00          | 9.78          | 10.67              |
>
> From the above results, we could see that even if the false font name only appears once in the sentence format, the VLMs are still easily to be cheated by the stroop effect, which demonstrates the issue again.
>
> We follow your suggestion to add the experiments to improve our paper.
>
> Again, thanks for your valuable feedback and meaningful suggestions!

---

> > ### Comment · Reviewer_A1JS · 2025-06-06
> >
> > Thank you for your response. I am still not satisfy with the current updates. Lack of open-source VLM's  results.

---

> > > ### Author Response · Authors · 2025-06-06
> > >
> > > We sincerely thank you for your continued thoughtful engagement with our work. We appreciate your suggestion and agree that testing on open-source VLMs is important for the completion of experiments.
> > >
> > > We follow your suggestion to test all open-source VLMs in the paper, the results are shown below:
> > >
> > > | Model Name | zero-shot | zero-shot CoT | mcq zero-shot | mcq zero-shot CoT |
> > > |----------|:---------:|:-------------:|:-------------:|:------------------:|
> > > | Llama-3.2-90B-Vision-Instruct | 9.78  | 8.00  | 10.67  | 10.22 |
> > > | Llama-3.2-11B-Vision-Instruct | 12.00 | 11.11 | 7.11 | 7.11 |
> > > | Phi-3.5-Vision-Instruct | 4.44  | 2.67 | 6.67 | 7.11 |
> > > | Phi-3-Vision-128k-Instruct | 2.22 | 0.89 | 6.67 | 5.78 |
> > > | Qwen2-VL-7B-Instruct | 5.78 | 4.44 | 7.11 | 6.67 |
> > > | Qwen2.5-VL-7B-Instruct | 8.00 | 7.11  | 8.00 | 7.56 |
> > > | Qwen2-VL-72B-Instruct | 8.44 | 9.33 | 8.89 | 9.78 |
> > > | Idefics3-8B-Llama3 | 4.44 | 2.67 | 6.67 | 6.67 |
> > > | Idefics2-8B | 9.33 | 7.56 | 6.67 | 8.00 |
> > >
> > > From the above results, we could reach the conclusion that open-source VLMs are also easily to be cheated by the stroop effect, which demonstrates the issue again. The performance of open-source VLMs does not appear to be significantly better than the experiment with single font name words, especially still around 6.67 under the setting of MCQ.
> > >
> > > Thank you again for your helpful suggestion and we follow your suggestion to add the additional experiments into our paper to enhance the conclusion.

---

### Comment · Program_Chairs · 2025-04-03

This paper violates the page limit due to adding a limitation sections beyond the page limit. COLM does not have a special provision to allow for an additional page for the limitations section. However, due to this misunderstanding being widespread, the PCs decided to show leniency this year only. Reviewers and ACs are asked to ignore any limitation section content that is beyond the 9 page limit. Authors cannot refer reviewers to this content during the discussion period, and they are not to expect this content to be read.

---

### Decision · Program_Chairs · 2025-07-08

**Decision:**

Accept

**Comment:**

Reviewers are somewhat split between borderline and positive. The paper presents an interesting effect of fonts on the performance of VLMs, as noted by at least two reviewers. Reviewers who are more negative had concerns about experiments that were in the opinion of the AC, fully satisfied for reviewer 1 (A1JS) but this is not reflected in the review score. Given that this is a somewhat unconventional topic, the AC sides more on the positive side for this work despite some reviewers' reservations.

[Automatically added comment] At least one review was discounted during the decision process due to quality]